# CompA: Addressing the Gap in Compositional Reasoning in Audio-Language Models

**Sreyan Ghosh**[♠][♦][#][*]    **Ashish Seth**[♠][*]    **Sonal Kumar**[♠][*]    **Utkarsh Tyagi**[♠][*]
**Chandra Kiran Evuru**[♠][*]    **S. Ramaneswaran**[♥]    **S. Sakshi**[♠]    **Oriol Nieto**[♦]
**Ramani Duraiswami**[♠]    **Dinesh Manocha**[♠]
[♠]University of Maryland, College Park, USA    [♦]Adobe, USA    [♥]NVIDIA, Bangalore, India
{sreyang, sonalkum, utkarsht, ckevuru, ramanid, dmanocha}@umd.edu

## ABSTRACT

A fundamental characteristic of audio is its compositional nature. Audio-language models (ALMs) trained using a contrastive approach (e.g., CLAP) that learns a shared representation between audio and language modalities have improved performance in many downstream applications, including zero-shot audio classification, audio retrieval, etc. However, the ability of these models to effectively perform compositional reasoning remains largely unexplored and necessitates additional research. In this paper, we propose **CompA**, a collection of two expert-annotated benchmarks with a majority of real-world audio samples, to evaluate compositional reasoning in ALMs. Our proposed CompA-order evaluates how well an ALM understands the order or occurrence of acoustic events in audio, and CompA-attribute evaluates attribute-binding of acoustic events. An instance from either benchmark consists of two audio-caption pairs, where both audios have the same acoustic events but with different compositions. An ALM is evaluated on how well it matches the right audio to the right caption. Using this benchmark, we first show that current ALMs perform only marginally better than random chance, thereby struggling with compositional reasoning. Next, we propose **CompA-CLAP**, where we fine-tune CLAP using a novel learning method to improve its compositional reasoning abilities. To train CompA-CLAP, we first propose improvements to contrastive training with composition-aware hard negatives, allowing for more focused training. Next, we propose a novel modular contrastive loss that helps the model learn fine-grained compositional understanding and overcomes the acute scarcity of openly available compositional audios. CompA-CLAP significantly improves over all our baseline models on the CompA benchmark, indicating its superior compositional reasoning capabilities.

## 1 INTRODUCTION

In recent years, multi-modal contrastive pre-training has emerged as an active area of research due to its great success in various downstream applications. CLIP (Radford et al., 2021), a pioneering vision-language model (VLM) trained using a multi-modal contrastive approach between vision and language representations, achieves state-of-the-art (SoTA) results in a variety of vision understanding tasks and enables crucial auxiliary tasks like image captioning (Mokady et al., 2021), text-to-image generation (Rombach et al., 2022), etc. Inspired by CLIP, Elizalde et al. (2023) introduce CLAP (Contrastive Language Audio Pre-Training), an audio-language model (ALM) that employs contrastive pre-training between audio and language representations and achieves SoTA in 16 downstream audio understanding tasks like zero-shot audio classification, text-to-audio retrieval, etc. CLAP imposes a shared representation space for audios and their text descriptions (captions), making it superior for audio-language perception than other models.

Despite much success, Wu et al. (2023) show that ALMs often act as bag of words and lack natural language comprehension. Understanding the relationship between text in captions and the corre-

---

*Equal technical contribution.[#]Work done as an intern at Adobe Research.

sponding content of the audio is a fundamental goal of audio processing, and the fact that different word orders correspond to differently perceived audio should be reflected in the capabilities of the ALMs. This phenomenon, also known as *compositional reasoning*, may be characterized as the ALM's capacity to understand the interrelationships among multiple discrete acoustic events in audio, such as order of occurence and attribute-binding, as conveyed through the words in the caption. Work in natural language has shown that transformers are often remarkably insensitive to word order (Sinha et al., 2021). Prior research has shown that models like CLIP, despite being trained on abundant data, exhibit a deficiency in compositional reasoning (Thrush et al., 2022; Ma et al., 2023; Yuksekgonul et al., 2023). Yuksekgonul et al. (2023) argue that one of the primary causes of this is that contrastive pre-training optimizes for retrieval, and it is easy for these models to perform well on retrieval even without having a good compositional understanding. In the past, researchers have proposed benchmarks (Thrush et al., 2022; Yuksekgonul et al., 2023) and tried inducing compositional reasoning in vision-language models (VLMs) (Yuksekgonul et al., 2023; Ma et al., 2023; Jiang et al., 2022), but no such attempt has been made in the audio space yet.

**Main Contributions.** In this paper, we perform the first systematic study for compositional reasoning in ALMs. To this end, we propose two novel expert-annotated benchmarks and a novel learning paradigm to teach ALMs compositional reasoning. Our main contributions are two-fold:

1. **We develop two expert-annotated benchmarks, CompA-order and CompA-attribute, with majority real-world audios, that serve as a test-bed to evaluate the compositional reasoning of ALMs.** CompA-order and CompA-attribute have 400 and 200 test instances, respectively, where each instance has two or three audio-caption pairs. Each audio is different in composition, and each caption contains the same words but in a different order to account for the different compositions. The task of an ALM is to correctly match the audios with their captions. While CompA-order is used to evaluate the models' ability to understand the order of occurrence between two acoustic events in an audio, CompA-attribute is used to evaluate the models' ability to link attributes to specific acoustic events (attribute-binding). More than 90% of audio snippets in CompA are sourced from real-world audio samples from AudioSet (Gemmeke et al., 2017) by expert annotators experienced in audio and language research. We discuss in Section 2 why current benchmarks do not evaluate compositional reasoning and how CompA serves as an essential step to fill this gap.

2. **We propose a robust learning solution for improving compositional reasoning in audio-language models**. First, we propose several improvements to contrastive learning with composition-aware hard negatives (Yuksekgonul et al., 2023). Employing hard negatives for each audio in the batch has proven to be an effective solution and we propose the following improvements: **(1)** We formulate the objective such that the hard negative captions for a particular audio are ignored by other audios in the batch. **(2)** We use an LLM to generate semantically viable negatives. Additionally, due to the lack of compositional audios in current training datasets, we propose a novel dataset with 110k+ audio-caption pairs based on the AudioSet strong subset (Hershey et al., 2021). Next, we propose a modular contrastive learning objective that does not require any existing compositional audio-caption pairs. Our proposed solution leverages a template-based audio-caption creation approach and improves CLAP's fine-grained attribute-binding and order-understanding capabilities. More specifically, we aim at aligning captions of various granularities to an audio, each of which represents a decomposed version of the audio scene. Additionally, we mine multiple negatives from the positives by interchanging orders and attributes. CompA-CLAP outperforms all our baselines on the CompA benchmark by 10%-28% while retaining performance on existing retrieval and zero-shot classification benchmarks. We open-source our code and data: `https://sreyan88.github.io/compa_iclr/`.

## 2 COMPA BENCHMARK

**Overview.** We propose two expert-annotated benchmarks, **CompA-order** and **CompA-attribute**, structured in the Winograd twin sentence format, to evaluate an ALMs' capabilities to understand various types of compositional relationships. Each instance in each benchmark has two or three audio-caption pairs, where each audio has the same acoustic events but with a different composition, and each caption has the same words (specifics detailed in next Sections). The task of an ALM

is to match the right caption with the right audio and vice-versa. We are particularly inspired by the Winoground dataset proposed by Thrush et al. (2022) built for evaluating visio-linguistic compositional reasoning. Our work is also inspired from Winograd twin sentence format, originally proposed in the Winograd schema challenge (Levesque et al., 2012), detailed in Section 5.

**Annotation.** Both benchmarks were annotated by four subject-matter experts specializing in audio and language. To build CompA-order, we used natural audio samples sourced from AudioSet Strong (Hershey et al., 2021). AudioSet Strong has temporal labels, and annotators were asked to annotate continuous stretches of audio with the desired acoustic events. We list down annotation rules, illustrate the snapshot of the annotation tool, and provide more information on the annotator backgrounds in Appendix B.3. We opted for natural audio over synthetic alternatives to capture inherent elements such as background noise and interventions, thereby offering a test-bed for ALMs that closely mimics real-world conditions. Finally, a continuous stretch of audio with the annotated events is sliced (with gold timestamps), and a caption is written for it.

## 2.1 WHY ARE CURRENT BENCHMARKS INSUFFICIENT FOR EVALUATING COMPOSITIONAL REASONING IN ALMS?

Fig. 1 illustrates the results of an experiment where we show that it is easy for ALMs to perform well on the most commonly used audio-retrieval benchmarks, Clotho (Drossos et al., 2020) and Audio-Caps (Kim et al., 2019), even without proper word ordering. Additionally, we analyze the distribution of unique nouns per caption and see that most of the audio samples in the test set of both benchmarks have only a single acoustic event and are non-compositional. Wu et al. (2023) also report that CLAP achieves similar benchmark performance when trained with captions stripped of all but nouns and verbs. This suggests that even ALMs without compositional reasoning abilities can perform well on these benchmarks.

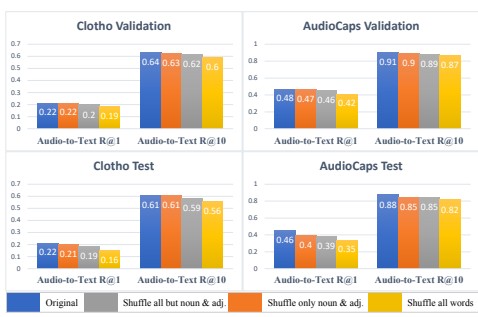

Figure 1: Comparison of CLAP performance after shuffling the word order in captions. CLAP undergoes an average degradation of 0.04 in top-1(R@1) and 0.03 in top-10 (R@10) retrieval.

## 2.2 COMPA-ORDER

Fig. 2 illustrates an example from CompA-order. We build CompA-order to evaluate an ALM's ability to understand the order of occurrence between multiple acoustic events. At its core, an acoustic event in an audio can either *succeed* another event, *precede* another event, or occur *simultaneously* with it. CompA-order has 400 test instances, with each instance comprising a minimum of two audio-caption pairs $(C_0,A_0;C_1,A_1)$, where the audios have the same two acoustic events, but the order of occurrence of the event in the audio differs. Out of the 400, 100 of these instances have an extra pair $(C_2,A_2)$, where the two acoustic events occur simul-

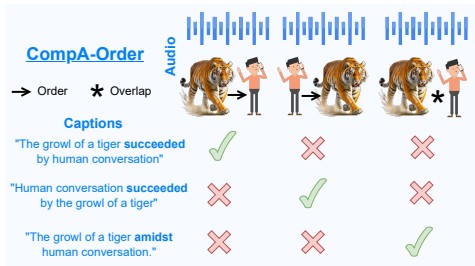

Figure 2: **CompA-order** evaluates an ALMs' capability to understand the *order of occurrence* between multiple acoustic events in an audio.

taneously. Expanding CompA with audios comprising more than two acoustic events remains part of future work. The captions for the audios in an instance with two pairs have the exact same words but in a different order, except for the instances with three pairs, where only a single word that defines the preposition between the events is changed. More details are in Appendix B.1.

## 2.3 COMPA-ATTRIBUTE

We build CompA-attribute to evaluate an ALM's ability to reason attribute-binding for acoustic events. CompA-attribute has 200 test instances, with each instance comprising two audio-caption pairs $(C_1,A_1;C_2,A_2)$, where each audio has the same two acoustic events

but as if associated with a different attribute. The term *attribute* is broad, and for our case, we consider multiple attributes like source (example in Fig. 3), qualitative ("Static hiss joins random notes played by a synthesizer." → "Static notes joins random hiss played by a synthesizer."), etc. The captions have the exact same words but in a different order. For CompA-attribute, we used synthetically generated audios from WavJourney Liu et al. (2023d) that are carefully validated by experts who also supervise the generation process. More details are in Appendix B.1 and B.3.

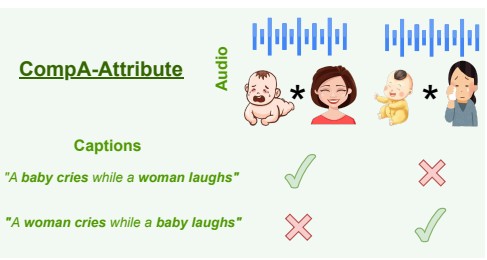

Figure 3: **CompA-attribute** evaluates an ALM's capability to understand *attribute-binding* for multiple acoustic events in an audio.

## 2.4 EVALUATION

Performance on both CompA-order and CompA-attribute is measured on three different metrics to measure different aspects of compositional reasoning in an ALM. We are inspired by Winoground (Thrush et al., 2022) for our evaluation metrics. Given two audios $A_0$ and $A_1$ and their corresponding captions $C_0$ and $C_1$, from an instance in either benchmark, we define the *text score* that measures whether an ALM can select the correct caption, given an audio. Thus, our metric $f(C_0, A_0, C_1, A_1)$ is defined as follows:

$$f(C_0, A_0, C_1, A_1) = \begin{cases} 1 & \text{if } s(C_0, A_0) > s(C_1, A_0) \\ & \text{and } s(C_1, A_1) > s(C_0, A_1) \\ 0 & \text{otherwise} \end{cases} \tag{1}$$

where $s(\cdot)$ is the cosine similarity between the audio-caption pair. Intuitively, if the model has a higher similarity for its ground-truth caption than the alternative caption describing a compositionally different audio, it performs better at compositional audio-to-text retrieval. The second metric is the *audio score*, which measures whether an ALM can select the correct audio, given a caption. We define our metric $g(C_0, A_0, C_1, A_1)$ as follows:

$$g(C_0, A_0, C_1, A_1) = \begin{cases} 1 & \text{if } s(C_0, A_0) > s(C_0, A_1) \\ & \text{and } s(C_1, A_1) > s(C_1, A_0) \\ 0 & \text{otherwise} \end{cases} \tag{2}$$

Intuitively, if the model has a higher similarity for its ground-truth audio than the alternative audio that is compositionally different, it performs better at compositional text-to-audio retrieval. Finally, we define a *group score* combining the audio and text scores defined above as follows:

$$h(C_0, A_0, C_1, A_1) = \begin{cases} 1 & \text{if } f(C_0, A_0, C_1, A_1) \\ & \text{and } g(C_0, A_0, C_1, A_1) \\ 0 & \text{otherwise} \end{cases} \tag{3}$$

The group score helps evaluate performance for an entire benchmark instance or set of sentences. This is motivated by prior work by Elazar et al. (2021), who show that the individual evaluation metrics tend to overestimate model performance by computing scores for the twin sentences individually. Finally, we average across all instances in the benchmark dataset for all three scores.

## 3 COMPA-CLAP: IMPROVING COMPOSITIONAL REASONING IN ALMS

Fig. 5 illustrates the training methodology of CompA-CLAP. CompA-CLAP is initialized with CLAP weights and is fine-tuned twice to learn compositionality. In the next sub-sections, we first discuss why currently available training datasets are insufficient to make an ALM learn compositional reasoning. Second, we detail the two stages of CompA-CLAP training, which prove to be superior to CLAP in compositional reasoning, when tested on CompA benchmarks.

## 3.1 WHY CURRENT TRAINING DATASETS ARE INSUFFICIENT TO LEARN COMPOSITIONAL REASONING? CAN SYNTHETIC DATA HELP?

**Problem.** Unlike vision-language models, which benefit from being trained on internet-scale data (Schuhmann et al., 2021), datasets for training ALMs are generally orders of magnitude smaller. While the presence of compositional audio in the training dataset does not guarantee the model will acquire compositional reasoning skills (Thrush et al., 2022), most proposed composition learning techniques from literature operate under the assumption that compositional audio is available within the training dataset (Yuksekgonul et al., 2023). The largest ALM to date is trained on only 630k audio-text pairs (Wu et al., 2023). The authors propose LAION-audio-630K, pooled from several existing openly available datasets. However, most of these audios do not have more than one acoustic event. Fig. 4 illustrates this phenomenon by show-

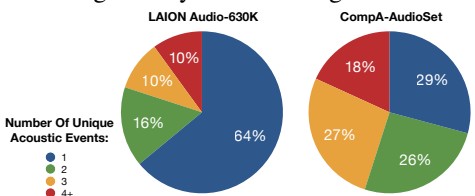

Figure 4: Comparison of unique acoustic events per audio between LAION Audio-630K and CompA-AudioSet.

ing the distribution of unique nouns per caption in LAION-audio-630K. To expand the pre-training dataset, the authors further propose keyword-to-caption augmentation on the large-scale AudioSet dataset (Gemmeke et al., 2017). Precisely, they feed the audio labels to a Pre-trained Language Model (PLM), which generates a coherent caption from the labels. However, we argue that such an augmentation scheme would generate captions that do not describe the correct ordering of events and would adversely affect the models' compositional reasoning abilities. Table 1 from Wu et al. (2023) also shows that augmenting LAIONAudio-630K with the entire 2M AudioSet for CLAP training almost never improves performance on any benchmark. We hypothesize that this is due to their Keyword-to-Caption generation algorithm, which does not guarantee temporally aligned captions.

**A simple yet not very effective solution.** A simple solution to the above-mentioned problem is to generate compositional audio using recent advancements in text-to-audio generation. We prompted SoTa text-to-audio generation models (Liu et al., 2023b; Ghosal et al., 2023), including WavJourney (Liu et al., 2023d) to generate audio for a caption generated using GPT-4. From our experiments, we conclude that current text-to-audio generation models struggle with diverse and compositional audio generation. Most of the time, the models struggled to generate cohesive audio with more than two acoustic events, and the generated audio did not constitute of events mentioned in the caption.

## 3.2 VANILLA CONTRASTIVE AUDIO-LANGUAGE PRE-TRAINING

**Methodology.** We train our own version of CLAP with contrastive audio-language pre-training but with minor modifications. For the audio encoder, we employ HTSAT-large(Chen et al., 2022), as also employed by Wu et al. (2023). For the text encoder, we replace the RoBERTa encoder with an instruction-tuned Flan-T5-large encoder (Chung et al., 2022). Ghosal et al. (2023) showed the effectiveness of employing an instruction-tuned encoder for multi-modal contrastive pre-training. Using our audio and text encoders, we solve the contrastive objective similar to CLAP (Elizalde et al., 2023), which solves the InfoNCE loss (Oord et al., 2018) between a batch of audio and text embeddings. Each audio in the batch of size $\mathcal{B}$ has one positive caption and $2\mathcal{B}$-1 negative captions. The same holds for text representations.

**Experimental Protocol.** For pre-training, we make minor modifications to the LAION-audio-630K pre-training dataset proposed by Wu et al. (2023). We introduce CompA-661k, with ≈661k unique audio-caption pairs. We list down all the sources of CompA-661k in Appendix B.2. Our version of CLAP outperforms (Wu et al., 2023) on all existing retrieval benchmarks for both text-to-audio and audio-to-text retrieval by 0.15%-4.67%, and CompA-order and CompA-attribute by 11.85%-23.8%.

## 3.3 CONTRASTIVE PRE-TRAINING WITH COMPOSITIONALLY-AWARE HARD NEGATIVES

**Overview & Background.** In this subsection, we describe our methodology to fine-tune an ALM using a modified contrastive learning (CL) formulation with compositionally-aware hard negatives. Following prior work in VLM fine-tuning (Yuksekgonul et al., 2023), we modify the vanilla CL objective and introduce compositionally aware hard negative captions for each audio in the batch to teach the model compositional reasoning. However, compositionally aware hard negative captions

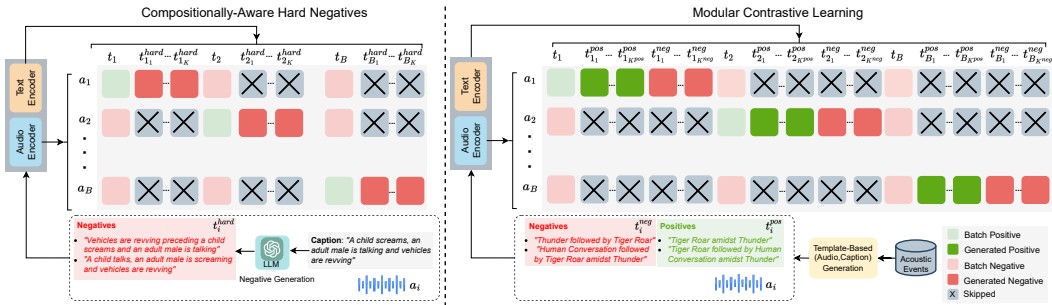

Figure 5: **Illustration of contrastive learning techniques for improving compositional reasoning in ALMs. Left:** Contrastive training with compositionally-aware hard negatives where each audio has $K$ hard negative captions generated using an LLM, and each audio in the batch ignores negatives of other audios in the batch for more focused training. **Right:** Our proposed Modular Contrastive training employs multiple positives and negatives for each audio in the batch generated using a template-based algorithm. Each positive describes compositional relationships of various granularities in the audio, and this helps the model learn fine-grained order and attribute-binding. An audio in the batch ignores the positives and negatives of other audios.

can only be formed for existing compositional audios, and the absence of it in open-source datasets is a considerable challenge to this approach.

**A new training dataset with compositionally rich audio-text pairs.** Building on the hypothesis that training on a compositionally rich audio dataset can lead to better compositional reasoning, we propose AudioSet-CompA, a new paired dataset, with ≈110k audio-text pairs for complex compositional audios. To build this dataset, we generated captions for the AudioSet strong by prompting an LLM and then asked two human annotators to evaluate and correct the captions. We experimented with both GPT-4 (OpenAI, 2023) and LLaMa-2 (Touvron et al., 2023), and we found GPT-4 to be better at generating captions. The AudioSet strong subset has temporally strong labels, i.e., it includes timestamps for custom prompt to generate captions that have the correct ordering of sound events. Fig. 4 compares the average number of acoustic events per audio in existing and our dataset.

**Methodology.** To mine compositionally aware hard negative captions from existing captions, we employ an LLM. Prior work in VLM fine-tuning employs traditional NLP techniques (Yuksekgonul et al., 2023), we found an LLM to better deal with anomalies and generate linguistically and semantically viable sentences. While most open- and closed-source LLMs are qualitatively competitive on this task, we adhere to GPT-4 as it performed better at generating structured outputs. A comparison of negatives generated using various LLMs and details on prompts used can be found in Appendix B.4. To generate hard negatives, we ask the LLM to swap acoustic event ordering, replace the preposition, or swap noun-verb associations. This helps generate negatives with similar contexts but different compositions. Additionally, we constrain the LLM to perform these operations only when the resultant swap resembles a plausible real-world scenario. This helps our model to learn more meaningful compositional relationships in the feature space. An example can be seen in Fig. 5, and more examples can be found in Appendix B.4.

Given a dataset $\mathcal{D}$ with $\mathcal{N}$ audio-caption pairs $(\mathcal{A}, \mathcal{T})$, for every audio $a_n \in \mathcal{A}$ and its corresponding caption $t_n \in \mathcal{T}$, we generate $K$ compositionally-aware negative captions $t_n^{\text{hard}}$. The generated negative captions serve as hard negatives in our modified contrastive fine-tuning objective. Unlike prior work, we formulate the objective such that hard negative captions for a particular audio are ignored by other audios in the batch. This helps in more composition-focused training. Thus, our final objective is defined as follows:

$$\ell_i^{t\text{-}2\text{-}a} = -t_i^\top a_i / \sigma + \log \sum_{j=1}^{B} \exp\left(t_i^\top a_j / \sigma\right) \tag{4}$$

$$\ell_i^{a\text{-}2\text{-}t} = -a_i^\top t_i / \sigma + \log\left(\sum_{j=1}^{B} \exp\left(a_i^\top t_j / \sigma\right) + \sum_{k=1}^{K} \exp\left(a_i^\top t_{i_k}^{\text{hard}} / \sigma\right)\right) \tag{5}$$

where $\ell^{t\text{-}2\text{-}a}$ is the contrastive loss for text that treats audio as negatives and $\ell^{a\text{-}2\text{-}t}$ is the contrastive loss for audio that treats text as negatives. $B$ is the batch size indexed by $i$ and $j$, $\sigma$

is the temperature parameter and $(t_{i_k}^{hard})_{k\in[1,K]}$ is the $k^{th}$ negative caption for audio sample $a_i$. Finally, we combine both the losses with appropriate scaling factors, $\alpha_1$ and $\alpha_2$, to optimize: $\mathcal{L}^{S_1} = \frac{1}{2B}\sum_{i=1}^{B}\left(\alpha_1 \ell_i^{t2a} + \alpha_2 \ell_i^{a2t}\right)$.

**Experimental Protocol.** For training, we use only the compositional audios from Clotho and AudioCaps in addition to our AudioSet-CompA dataset. This results in ≈100k training pairs, which is significantly low-resource compared with prior-art teaching a model compositional reasoning. We use $K{=}3$ negatives and provide values for other values of $K$ in Appendix C.1. We initialize the training with vanilla CLAP weights and only train the last few layers to overcome the problem where fine-tuning distorts pre-trained features when the fine-tuning and pre-training algorithms don't match (Wortsman et al., 2022; Goyal et al., 2023). Hyper-parameters details in Appendix B.7.

## 3.4 MODULAR CONTRASTIVE LEARNING FOR FINE-GRAINED COMPOSITION UNDERSTANDING

**Overview & Background.** This Subsection describes modular contrastive learning, a novel contrastive learning formulation backed with a novel synthetic data creation methodology that overcomes problems in compositionally-aware hard negative contrastive training, like scarcity of existing compositional audio-caption pairs and learning fine-grained compositional reasoning.

As discussed earlier, ALM training suffers from an acute lack of compositional audio. Thus, scaling contrastive pre-training with hard negatives poses a significant challenge. Additionally, a single hard negative for an audio with multiple difficult-to-distinguish acoustic events can be too complicated for the model to understand effectively. This is because real-world audio is often complex and unpredictable. This hinders the model from learning fine-grained attribute-binding and order information. Thus, we propose a new learning methodology that can be easily scaled, does not require any existing compositional audio-caption pairs, and helps the model learn fine-grained compositional reasoning. More specifically, we employ a modular template-based approach to create compositional audios and their captions from single acoustic events and their labels (which are abundantly and easily available). Then, we align each audio in the batch with positive captions of various granularity and generate hard negatives with different compositions, which allows the model to focus on fine-grained compositional relationships in the audio.

**Template-based synthetic creation of audio-caption pairs.** Fig. 6 illustrates the proposed algorithm. We propose a simple and scalable template-based approach to create compositional audio, their caption, and hard negatives for training. We first create a pool of several unique audio snippets and their labels, each comprising a single acoustic event. Each label might have several unique audio snippets corresponding to it. Using the snippet and the label corresponding to the acoustic event of the audio, we either concatenate these audios or overlay one audio over the other (More details in B.6.1). For this paper, we slice longer audios from the AudioSet strong subset (with the provided temporal labels) to create the pool of unique audio snippets and employ a maximum of 4 snippets corresponding to 4 unique events for synthetic audio creation. Additionally, to assure high quality, we don't concatenate or overlay random acoustic events but ask an LLM to create unique audio scenes based on the available labels (examples in Table 9). To create its corresponding caption,

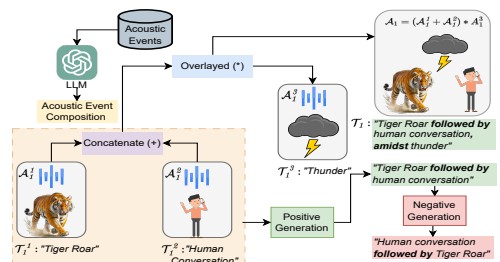

Figure 6: **Illustration of our template-based audio-caption creation process.** An LLM first generates a scene from a pool of available acoustic events. We then perform simple operations to generate compositional audio and their captions. This process also enables us to generate positives (and their corresponding negatives) that define fine-grained relations between events.

we employ pre-defined templates with diverse prepositions. Additionally, since we create compositional audio in a modular fashion, it allows us to easily obtain captions of various granularity and complexities, each defining one or multiple compositional relationships in the audio. To generate a hard negative that defines an alternate composition, we either switch attributes among events or word order. Examples can be found in Table 7.

Table 1: Performance comparison of CompA-CLAP with baselines on benchmark datasets. **Left:** Text-to-Audio and Audio-to-Text retrieval performance on AudioCap / Clotho (in the same format). **Right:** Zero-shot classification results on four benchmark datasets. While our CLAP achieves SoTA performance in almost all cases, CompA-CLAP retains its performance even after fine-tuning for compositionality.

| Model | T-A Retrieval | | | A-T Retrieval | | | | ESC-50 | US8K | VGGSound | FSD50K |
|---|---|---|---|---|---|---|---|---|---|---|---|
| | R@1 | R@5 | R@10 | R@1 | R@5 | R@10 | | | | | |
| MMT | 36.1 / 6.7 | 72.0 / 21.6 | 84.5 / 33.2 | 39.6 / 7.0 | 76.8 / 22.7 | 86.7 / 34.6 | Wav2CLIP | 41.4 | 40.4 | 10.0 | 43.1 |
| ML-ACT | 33.9 / 14.4 | 69.7 / 36.6 | 82.6 / 49.9 | 39.4 / 16.2 | 72.0 / 37.6 | 83.9 / 50.2 | AudioClip | 69.4 | 65.3 | - | - |
| CLAP | 34.6 / 16.7 | 70.2 / 41.1 | 82.0 / 54.1 | 41.9 / 20.0 | 73.1 / 44.9 | 84.6 / 58.7 | CLAP | 82.6 | 73.2 | - | 58.6 |
| CLAP-LAION | **36.2 / 17.2** | 70.3 / 42.9 | 82.5 / 55.4 | 45.0 / **24.2** | 76.7 / 51.1 | 88.0 / 66.9 | CLAP-LAION-audio-630K | 88.0 | 75.8 | 26.3 | 64.4 |
| CLAP (*ours*) | 35.9 / 17.0 | 78.3 / **44.1** | 89.6 / **56.9** | 47.8 / 23.8 | 83.2 / **51.8** | 90.7 / **67.8** | CLAP (*ours*) | **90.2** | **86.1** | 29.1 | **77.8** |
| CompA-CLAP (*ours*) | 36.1 / 16.8 | **78.6** / 43.5 | **90.2** / 56.1 | **47.8** / 23.9 | **83.5** / 50.7 | 90.2 / 67.6 | CompA-CLAP (*ours*) | 89.1 | 85.7 | **29.5** | 77.4 |

**Methodology.** Given a pool of acoustic events and their labels $\mathcal{P}$, we first generate a dataset $\mathcal{D}$ with $\mathcal{N}$ training instances, where each instance consists of an audio, its fine-grained positive captions, and their corresponding negatives denoted as $(\mathcal{A}, \mathcal{T}^{pos}, \mathcal{T}^{neg})$. Every instance $t_i^{pos} \in \mathcal{T}^{pos}$ is a set of $K^{pos}$ fine-grained positives, denoted by $t_{i_{k \in K^{pos}}}^{pos}$, and similarly $t_i^{neg} \in \mathcal{T}^{neg}$ is a set of $K^{neg}$ fine-grained negatives, denoted by $t_{i_{k \in K^{neg}}}^{neg}$. Other audios in the batch ignore the generated fine-grained positives and negatives for a particular audio. Thus, we define our objective as follows:

$$\ell_i^{t\text{-}2\text{-}a} = -\left( \frac{1}{K^{pos}} \sum_{k=1}^{K^{pos}} (t_{i_k}^{pos})^\top a_i / \sigma \right) + \log \sum_{j=1}^{B} \exp\left( t_i^\top a_j / \sigma \right) \tag{6}$$

$$\ell_i^{a\text{-}2\text{-}t} = -\left( \frac{1}{K^{pos}} \sum_{k=1}^{K^{pos}} a_i^\top t_{i_k}^{pos} / \sigma \right) + \log\left( \sum_{j=1}^{B} \exp\left( a_i^\top t_j / \sigma \right) + \sum_{k=1}^{K^{neg}} \exp\left( a_i^\top t_{i_k}^{neg} / \sigma \right) \right) \tag{7}$$

where $\ell^{t\text{-}2\text{-}a}$ is the contrastive loss for text that treats audio as negatives and $\ell^{a\text{-}2\text{-}t}$ is the contrastive loss for audio that treats text as negatives. $B$ is the batch size indexed by $i$ and $j$, $\sigma$ is the temperature parameter. $(t_{i_k}^{pos})_{k \in [1, K^{pos}]}$ and $(t_{i_k}^{neg})_{k \in [1, K^{neg}]}$ are $k^{th}$ generated fine-grained positive and negative caption for audio sample $a_i$. Finally we combine both the losses with appropriate scaling factors $\beta_1$ and $\beta_2$, to optimize: $\mathcal{L}^{S_2} = \frac{1}{2B} \sum_{i=1}^{B} \left( \beta_1 \ell_i^{t\text{-}2\text{-}a} + \beta_2 \ell_i^{a\text{-}2\text{-}t} \right)$.

**Experimental Protocol.** We generate $\approx$251k audios for training. Our pool $\mathcal{P}$ has $\approx$500k unique audio snippets. The maximum number of acoustic events we use to make an audio is 4. Since the number of positives and negatives for each audio can grow combinatorially, we restrict the maximum to 7 for each. We initialize our model from CLAP, fine-tuned with hard negatives. Training hyperparameters are detailed in Appendix B.7. We call the resultant model CompA-CLAP.

## 4 RESULTS

Table 1 (left) compares CompA-CLAP with four other baselines on text-to-audio and audio-to-text retrieval tasks on Clotho and AudioCaps. These four baselines are MMT (Oncescu et al., 2021), ML-ACT (Mei et al., 2022), CLAP (Elizalde et al., 2023) and CLAP-LAION (Wu et al., 2023). While our CLAP trained on CompA-661k outperforms all baselines in most cases, CompA-CLAP, fine-tuned for better compositional reasoning, performs

Table 2: Performance comparison of CompA-CLAP with baselines on CompA-order and CompA-attribute.

| Model | CompA-order | | | CompA-attribute | | |
|---|---|---|---|---|---|---|
| | Text | Audio | Group | Text | Audio | Group |
| Human | 90.60 | 91.20 | 87.40 | 80.30 | 82.40 | 79.80 |
| Random | 19.70 | 19.70 | 16.67 | 25.0 | 25.0 | 16.67 |
| MMT | $19.90_{\pm1.30}$ | $6.85_{\pm1.90}$ | $3.90_{\pm1.95}$ | $29.59_{\pm1.03}$ | $4.69_{\pm2.29}$ | $3.12_{\pm1.76}$ |
| ML-ACT | $21.85_{\pm1.75}$ | $8.00_{\pm0.80}$ | $4.35_{\pm1.25}$ | $31.63_{\pm1.46}$ | $5.11_{\pm2.02}$ | $3.75_{\pm0.86}$ |
| CLAP | $22.80_{\pm2.15}$ | $8.35_{\pm1.40}$ | $4.70_{\pm2.20}$ | $33.27_{\pm0.72}$ | $6.14_{\pm1.37}$ | $4.66_{\pm2.08}$ |
| CLAP-LAION | $24.0_{\pm1.10}$ | $9.25_{\pm1.15}$ | $5.50_{\pm0.80}$ | $34.78_{\pm1.45}$ | $6.52_{\pm1.47}$ | $5.07_{\pm1.62}$ |
| CompA-CLAP (*ours*) | $40.70_{\pm0.10}$ | $35.60_{\pm0.15}$ | $33.85_{\pm0.15}$ | $44.28_{\pm0.07}$ | $22.52_{\pm0.06}$ | $15.13_{\pm0.09}$ |
| - Hard Negative | $36.25_{\pm0.15}$ | $31.45_{\pm0.10}$ | $20.20_{\pm0.05}$ | $39.27_{\pm0.17}$ | $17.71_{\pm0.13}$ | $11.35_{\pm0.19}$ |
| - Modular Contrastive | $38.0_{\pm0.15}$ | $33.50_{\pm0.20}$ | $21.25_{\pm0.10}$ | $43.48_{\pm0.11}$ | $19.57_{\pm0.16}$ | $13.04_{\pm0.21}$ |
| CLAP (*ours*) | $33.75_{\pm0.05}$ | $15.75_{\pm0.15}$ | $11.50_{\pm0.15}$ | $42.40_{\pm0.07}$ | $20.50_{\pm0.05}$ | $14.75_{\pm0.13}$ |

on par with CLAP with minimal performance degradation. Table 1 (right) compares CompA-CLAP with four other baselines on zero-shot classification datasets on ESC-50 (Piczak), US8K (Salamon et al., 2014), VGGSound (Chen et al., 2020) and FSD50k (Fonseca et al., 2022). The four baselines are Wav2CLIP (Wu et al., 2022), AudioCLIP (Wu et al., 2022), CLAP, and CLAP-LAION. All scores have been averaged for 3 runs on 3 random seeds.

Table 2 compares the results of CompA-CLAP with our baselines on CompA-order and CompA-attribute benchmarks. First, our vanilla CLAP performs better than all other baselines from literature, outperforming CLAP-LAION by $\approx$6%-33% over both benchmarks. CompA-CLAP, which is

CLAP trained consecutively with hard negatives and modular contrastive learning, improves performance on both benchmarks by ≈10%-28% over CLAP. We also notice ≈4%-13% performance degradation when compositionally-aware hard negative training is skipped before Modular contrastive training. Modular contrastive training after hard negative training improves performance on benchmarks by ≈2%-11%. However, all models, including CompA-CLAP, perform worse than our random baseline on CompA-attribute, which leaves plenty of room for improvement.

We also discuss some common mistakes observed upon result analysis. 1) CompA-CLAP performs better in cases when the acoustic events in an audio sound more distinct (eg., tiger growling and human speaking) and underperforms where they sound very similar (eg., human sounds). 2) CompA-CLAP also suffers from the long-tailed problem in AudioSet, wherein it underperforms in audios with acoustic events seen less during training. 3) In CompA-order, the model underperforms on prepositions not seen during training.

## 5 RELATED WORK

**Compositional Reasoning.** Early work in linguistics has tried to understand what models know about word order (Sinha et al., 2021), syntax (Gauthier et al., 2020; Gulordava et al., 2018; Hu et al., 2020; Linzen et al., 2016), or the complex interaction between syntactic and semantic categories (Kann et al., 2019; Thrush et al., 2020; Warstadt et al., 2019; 2020). Learning visio-linguistic compositionality has been extensively studied in prior art (Yuksekgonul et al., 2023; Ma et al., 2023). Winground (Thrush et al., 2022), the work closest to ours, proposes a benchmark with 400 test cases in the Winograd twin sentence format, with pairs of compositionally different images and captions with the same words but in a different order. The task, similar to the one proposed in this paper, is to match the right image with the right caption. They additionally show that current VLMs perform no better than random chance. The Winograd twin sentence format was originally proposed in the Winograd schema challenge (Levesque et al., 2012) and has been earlier used for a variety of language-related tasks (Rudinger et al., 2018; Sakaguchi et al., 2021; Zhao et al., 2018). Following this, (Yuksekgonul et al., 2023) propose a large-scale benchmark with over 50,000 test cases and compositionally different captions by swapping relational tokens within sentences. Lack of compositional reasoning in VLMs has affected multiple downstream tasks like text-to-image generation (Conwell & Ullman, 2022) and Visual Question Answering (Bogin et al., 2021). A similar problem was observed in AudioLDM (Liu et al., 2023a), which employs CLAP as a text encoder for text-to-audio generation and fails to generate compositional audios (Ghosal et al., 2023).

**Audio and Language.** Recent developments indicate an increasing trend in leveraging language as a modality for interaction with audio systems. Downstream tasks like text-to-audio generation (Ghosal et al., 2023; Liu et al., 2023a; Huang et al., 2023) and text-to-music generation (Agostinelli et al., 2023) have gained much popularity. Other tasks include text-guided audio source separation (Liu et al., 2023c) audio captioning (Ghosh et al., 2023), etc. Deshmukh et al. (2023) integrate language models with audio encoders and frame all audio tasks as text-generation tasks. Their model, Pengi, achieves SoTA performance on 22 downstream tasks, which shows promises of effective language modality integration for enhanced audio system interactions. Most of these models employ a text encoder or audio encoder to accomplish their task. CLAP, which learns a shared representation between audio and language modalities and achieves impressive performance on zero-shot tasks, proving to be a compelling model for cross-modal understanding and interaction.

## 6 CONCLUSION AND FUTURE WORK

In this paper, we first discuss several problems with existing audio-retrieval benchmarks and show that it is easy for ALMs to perform well on these benchmarks without any compositional understanding. To address this gap, we propose CompA, a suite of two benchmarks to evaluate an ALM's compositional reasoning capabilities. Using this benchmark, we first show that current ALMs struggle with compositional reasoning. Next, we propose CompA-CLAP, an audio-language model that addresses the lack of compositional reasoning in ALMs. To train CompA-CLAP, we first propose improvements to contrastive pre-training with compositionally-aware hard negatives. Then, we propose a novel modular contrastive training approach that helps the model learn fine-grained order information and attribute binding. As part of future work, we would like to expand the size of CompA and find better and novel solutions to teach an ALM compositional reasoning.

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

## A  COMPOSITIONALITY

Compositionality refers to the property that the meaning of a complex expression is determined by the meanings of its constituent parts and the rules used to combine them. In the context of audio, compositionality can manifest in various ways. Below are different types of compositionalities observed in audio:

1. **Temporal Compositionality**: Temporal compositionality refers to the organization and combination of audio elements over time. It involves how individual sounds or musical notes are arranged and structured to form a cohesive and meaningful audio sequence. Examples include rhythms, melodies, and the timing of sound events within a composition.

2. **Attribute Compositionality**: Attribute compositionality in audio refers to the manner in which different perceptual or structural attributes of sound elements combine or interact to create a complex auditory experience.

3. **Spectral Compositionality**: Spectral compositionality pertains to the combination of frequency components within an audio signal. It involves the arrangement of different frequencies and how they interact to create the overall timbre or color of the sound. In music, this can relate to the orchestration of different instruments or the layering of various sonic elements.

4. **Harmonic Compositionality**: Harmonic compositionality focuses on the relationships between different pitches or tones in an audio composition. It involves the creation of harmonies, chords, and the interplay between different musical notes to produce a harmonically rich and structured auditory experience.

5. **Spatial Compositionality**: Spatial compositionality refers to the arrangement and organization of sound sources in space. In the context of audio production or immersive audio experiences, it involves how sounds are positioned, panned, or spatially distributed to create a sense of depth, width, and localization within the auditory space.

6. **Timbral Compositionality**: Timbral compositionality relates to the manipulation and combination of different timbres or sonic qualities. It involves how variations in tone color, texture, and sound processing contribute to the overall expressive and aesthetic qualities of the audio. Timbral compositionality is often prominent in electronic music and sound design.

7. **Semantic Compositionality**: Semantic compositionality extends beyond the perceptual aspects of sound and encompasses the symbolic or semantic meaning associated with audio elements. It involves how sounds convey meaning, emotions, or convey specific messages within a larger audio context.

## B  ADDITIONAL DETAILS

### B.1  COMPA BENCHMARK

Table 3: **Left:** Frequency of different kinds of sounds in the CompA-order benchmarks. The grouping was done manually by the human annotators (top). **Right:** Frequency of unique parts-of-speech in the audio captions.

| Description | #CompA-order | #CompA-attribute |
|---|---|---|
| Human Speech | 505 | 166 |
| Background Sound | 210 | 10 |
| Bird Sounds | 151 | 5 |
| Land Animal Sounds | 144 | 20 |
| Background Sound Distinct | 142 | 8 |
| Human Sound | 136 | 188 |
| Alarm and Siren | 120 | 18 |
| Household and Indoor Sounds | 116 | 7 |
| Mechanical and Electrical Sounds | 112 | 39 |
| Bell Sound | 62 | 4 |
| Tools and Equipment Sound | 41 | 16 |
| Vehicle Sound | 40 | 32 |
| Water Sound | 34 | 14 |
| Mobile and Telephone Sound | 26 | 0 |
| Wind Sound | 25 | 13 |
| Construction Sound | 22 | 4 |

| | CompA-Order | CompA-Attribute |
|---|---|---|
| Unique Nouns | 795 | 296 |
| Unique Verbs | 250 | 163 |
| Unique Prepositions | 21 | 27 |
| Unique Adjectives | 99 | 78 |

Table 3 (left) shows the frequency distribution of various kinds of sounds among all instances in the CompA-order and CompA-attribute benchmarks. Both benchmarks benefit from acoustic diversity. For CompA-order, the highest occurrence includes human speech and background sounds, and the lowest occurrence of wind and construction sounds. For CompA-attribute, the highest occurrence includes human speech and human sound, and the lowest occurrence of mobile and telephone sounds and construction sounds. Human Speech consists of audios of *female speech, woman speaking, child speech, etc*, Background Sound consists of audios of *breaking, rumbling, etc*, Bird Sound consists of audios of *chirping, quacking, etc*, Land Animal Sound consists of audios of *roar, moo, etc*, Background Sound Distinct consists of audios of *revving, explosion, etc*, Human Sound consists of audios of *sneezing, cheering, etc*, Alarm and Siren consists of audios of *steam Whistle, alarm clock, etc*, Household and Indoor Sound consists of audios of *chopping, scraping, etc*, Mechanical and Electrical Sound consists of audios of *typewriter, clicking, etc*, Bell Sound consists of audios of *door Bell, church bell, etc*, Tools and Equipment Sound consists of audios of *gunshot, electric razor, etc*, Vehicle Sound consists of audios of *train horn, motorcycle, etc*, Water Sound consists of audios of *rain, waves, etc*, Mobile and Telephone Sound consists of audios of *telephone ringing, dial tone, etc*, Wind Sound consists of audios of *wind chime, wind howl, etc*, Construction Sound consists of audios of *drilling, hammering, etc*.

Tab 3 (right) shows the frequency of various parts of speech in CompA-order and CompA-attribute. Both benchmarks benefit from rich linguistic diversity. This emphasizes a more robust evaluation of ALMs with our benchmark.

Table 6 provides examples from both benchmarks.

### B.2  COMPA-661K

Table 4 lists all sources from which CompA-661k is pooled. We propose several changes over LAION-audio-630K.

Table 4: List of sources from where CompA-661K is pooled.

| Dataset | #Sents |
|---|---|
| MACS (Morato & Mesaros, 2021) | 14400 |
| ESC-50 (Piczak) | 2000 |
| SONISS (Sonniss Limited, 2022) | 1631 |
| Musical Instrument (Agostinelli et al., 2023) | 9774 |
| SoundBible (sou, 2023) | 1232 |
| LibriTTS (Zen et al., 2019) | 93772 |
| Free Sound (Fonseca et al., 2022) | 259020 |
| Medley-solos (Lostanlen et al., 2019) | 732 |
| WavText5K (Deshmukh et al., 2022b) | 4347 |
| MusicCaps (Agostinelli et al., 2023) | 2645 |
| GTZAN (Tzanetakis et al., 2001) | 6014 |
| FSD50K (Fonseca et al., 2022) | 40966 |
| BBC (BBC, 2018) | 31201 |
| Urbansound8K (Salamon et al., 2014) | 17410 |
| CompA-AudioSet (ours) | 108311 |
| AudioCaps (Kim et al., 2019) | 48649 |
| Clotho (Drossos et al., 2020) | 19195 |

Table 5: List of various datasets used in Pre-training and Evaluation

| Dataset | Source | Experiment |
|---|---|---|
| **Pre-training** | | |
| CompA-661K | refer to Table 4 | Vanilla Contrastive Learning |
| CompA-AudioSet | AudioSet (Gemmeke et al., 2017) | Contrastive Learning with Hard Negatives |
| Modular Contrastive Dataset | AudioSet Strong (Hershey et al., 2021) | Modular Contrastive Learning |
| **Evaluation** | | |
| CompA-order | AudioSet (Gemmeke et al., 2017) | Evaluation Dataset |
| CompA-attribute | AudioSet (Gemmeke et al., 2017) WavJourney (Liu et al., 2023d) | Evaluation Dataset |

Table 6: Examples from the CompA-order (left) and CompA-attribute (right) benchmark.

| Description | CompA-order Captions |
|---|---|
| Caption | The growl of a tiger succeeded by human conversation. |
| Rev. Caption | Human conversation succeeded by the growl of a tiger. |
| Triplet | The growl of a tiger amidst human conversation. |
| Caption | A barking dog preceding a man's conversation. |
| Rev. Caption | A man's conversation preceding a barking dog. |
| Triplet | A man conversing with a barking dog in the background. |
| Caption | A howling dog, soon after joined by a speaking woman. |
| Rev. Caption | A speaking woman, soon after joined by a howling dog. |
| Triplet | A howling dog intermixed with a speaking woman. |
| Caption | A bleating goat followed by people speaking. |
| Rev. Caption | People speaking. followed by a bleating goat. |
| Triplet | People speaking intermingled with a bleating goat. |
| Caption | A conversation occurring prior to the telephone chime. |
| Rev. Caption | The telephone chime occurring prior to a conversation. |
| Triplet | A conversation interleaved with a telephone chime. |
| Caption | A man speaking prior to the usage of a cash register. |
| Rev. Caption | A cash register being used prior to a man speaking. |
| Triplet | A man speaking while a cash register is being used. |

| Description | CompA-attribute Captions |
|---|---|
| Caption | A crowd speaks and a man cheers. |
| Rev. Caption | A crowd cheers and a man speaks. |
| Caption | As a woman whistles, a young kid makes a declaration and a person speaks and a young male shouts out excitedly |
| Rev. Caption | As a woman speaks, a young kid make a declaration and a person whistles and a young male shouts out excitedly |
| Caption | Police noise blares amid general town siren |
| Rev. Caption | Police siren blares amid general town noise |
| Caption | Chirping with light wind and distant rustling |
| Rev. Caption | Rustling with light wind and distant chirping |
| Caption | Light music buzzing followed by static electricity briefly playing as rain falls on a surface. |
| Rev. Caption | Light music playing followed by static electricity briefly buzzing as rain falls on a surface |
| Caption | A music box ticks a tune and a clock plays |
| Rev. Caption | A music box plays a tune and a clock ticks |
| Caption | A baby talking over a radio as a man is crying |
| Rev. Caption | A baby crying over a radio as a man is talking |

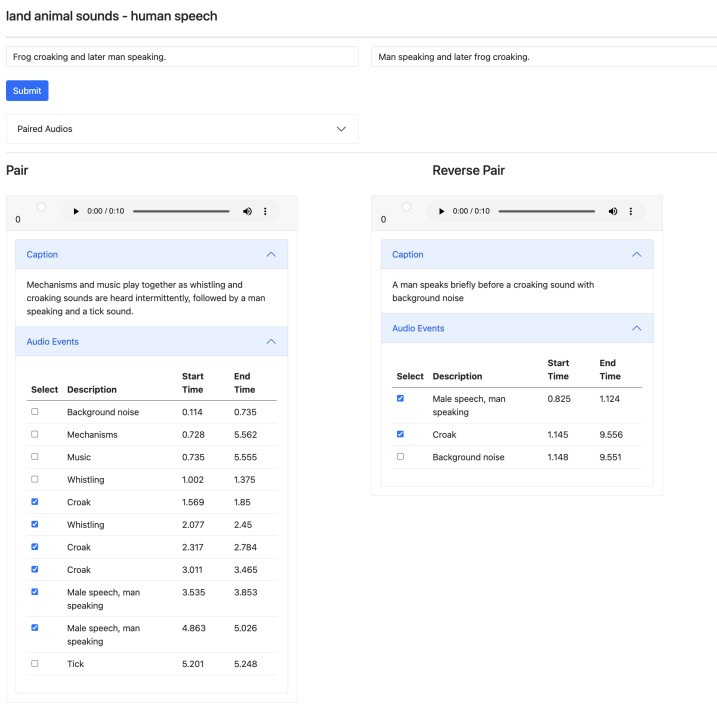

Figure 7: Snapshot of the illustration tool used for annotating CompA-order.

### B.3 ANNOTATION

Fig. 7 illustrates a snapshot of the annotation tool used for CompA-order. The audio and the corresponding timestamps are obtained from AudioSet strong subset (Hershey et al., 2021). We first perform pre-processing to bring down the search space of paired audios. Details about pre-processing can be found later in the Section. Next, two expert annotators find all possible pairs of audio where two acoustic events occur in the opposite order using an annotation tool illustrated in Fig. 7. Natural interventions or overlayed with natural disturbances were allowed. Additionally, each annotator writes creative captions for the identified audio segment. Captions for both audios are instructed to have the same words but different word ordering to account for the compositional change. Post the annotators found 400 such pairs of audio-caption pairs, they were instructed to find a triplet audio segment for any 100 such pairs from a pool of available audio segments, where the audio events in the pair occurred simultaneously with each other.

For CompA-attribute, the entire process was done manually. In the first stage, only the audios where an attribute swap resulted in genuine real-world audio were selected. Then, for each audio, an audio with swapped attributes was found manually. Finally, if no such audio was found, Wavjourney (Liu et al., 2023d) was employed to generate the audio. Post this step, a manual inspection was done to validate the clarity of the events in the audio.

**Annotator Demographics.**Our team of annotators comprises highly skilled domain experts in audio, each bringing a wealth of research experience. They possess a deep understanding of sound analysis and are adept at discerning intricate details in audio recordings. Their expertise is not just technical but also theoretical, allowing them to approach the annotation process with a nuanced perspective. This background enables them to handle complex audio data with precision and insight, ensuring high-quality annotations that are both accurate and meaningful. Their combined experience in audio research is a valuable asset to our project, contributing significantly to the depth and reliability of our annotated audio corpus.

**Annotator Process.** Out of the total 600 pairs in both benchmarks, all four annotators annotated 150 pairs each (100 and 50 for CompA-order and CompA-attribute, respectively). After the initial

round of annotations, each annotator reviewed and annotated a set previously handled by a fellow annotator in a second round (assigned randomly).

**Annotation Rules.**

1. **Primary Event Focus:** Annotators must prioritize the identification and labeling of the two main events in each audio. While acknowledging other background sounds, the focus should be on clearly distinguishing and documenting these primary events.

2. **Background Sound Limitation:** Background or incidental sounds can be present, but they should not overpower or obscure the primary events. Annotators should ensure that these main events remain distinct and easily identifiable despite the presence of additional sounds.

3. **Consistent Event Ordering:** Each audio contains the same two events, but in reverse order. Annotators should carefully document the sequence of these events, ensuring accuracy in reflecting their reversed order across paired audios.

4. **Clarity and Distinctiveness:** In instances where natural sounds or other auditory elements are present, annotators should verify that these do not interfere with the clarity and distinctiveness of the main events. The two primary events should be identifiable without ambiguity.

5. **Contextual Relevance:** Any additional sounds included in the recordings should be contextually relevant and not misleading. These sounds should support the primary events' setting or scenario without leading to confusion about the main events themselves.

**Annotation Pre-processing.** To decrease the search space of finding pairs of audios with the desired acoustic events and opposite order of occurrence, we filter the search space using rules and logic. We provide the code in our Supplementary.

### B.4 COMPARISON OF VARIOUS TECHNIQUES FOR NEGATIVE MINING

Table 7 compares various LLMs and traditional NLP techniques for mining compositionally aware hard negatives for contrastive training. While LLaMa (Touvron et al., 2023) and LLaMa-2 (Touvron et al., 2023) are competitive in quality to GPT-4, Falcon (Penedo et al., 2023) almost always produces the same output as the input caption. On the other hand, traditional NLP techniques using spacy (Honnibal & Montani, 2017) almost always generate sentences that are not linguistically and semantically viable.

### B.5 PROMPTS

We use two prompts to generate the negatives of a given caption - fist to generate the negatives by changing the temporal order of events in the caption, and second to generate the negatives by swapping the verbs associated with the nouns until the the sentence made sense after swapping. The prompt used for both are listed below respectively:

1. I will provide you with a list. Each item in the list is a caption of an audio file. The caption may have a temporal sequence of multiple audio events taking place. The temporal sequence of events can be of any order, which may describe sounds occurring together, followed by each other, etc. You have to return 3 modified captions that change the audio composition by changing the temporal order of events through modifying phrases in the caption. Except the words that contribute to the ordering of the events, the wording of the actual events should be kept as unchanged as possible. You are only allowed to make minor modifications to words that do not contribute to the temporal sequence, so that the modified caption is linguistically correct. Return only a list for each caption in the input list, with compositionaly modified caption. Here are some examples: Original Caption: 'A man speaks twice before an electric toothbrush starts, followed by clicking sounds.', Modified Captions: ['Clicking sounds occur after a man speaks twice, followed by the start of an electric toothbrush..', 'A man speaks twice with an electric toothbrush starting and clicking sounds in background', 'A man speaks twice after an electric toothbrush starts amidst clicking sounds in background', 'A man speaks twice as clicking sounds occur, followed

Table 7: Comparison of different LLMs and Traditional NLP technique to generate negative of the given Original caption

| Model | Caption |
|---|---|
| Original | Someone sneezes, a crowd laughs |
| GPT-4 | Someone laughs, a crowd sneezes |
| LLaMa | creaming is heard, followed by a large gun being fired. |
| LLaMa-2 | A crowd laughs, followed by someone sneezing. |
| Falcon | Someone laughs, a crowd sneezes |
| Traditional NLP | crowd sneezes , a Someone laughs |
| Original | An infant crying as a man is talking followed by a toy train horn honking |
| GPT-4 | An infant talking as a man is crying followed by a toy train horn honking |
| LLaMa | A man is talking and an infant cries, amidst the sound of a toy train horn honking. |
| LLaMa-2 | A man is talking and an infant cries, amidst the sound of a toy train horn honking. |
| Falcon | An infant talking as a man is crying followed by a toy train horn honking |
| Traditional NLP | An infant train as a crying is talking followed by a honking man toy horn |
| Original | A woman speaks and then a kid yells loudly as soft voices speak softly in the background |
| GPT-4 | A woman yells and then a kid speaks loudly as soft voices speak softly in the background |
| LLaMa | Soft voices speak softly in the background, then a woman speaks, followed by a kid yelling loudly. |
| LLaMa-2 | Soft voices speak softly in the background, then a woman speaks, followed by a kid yelling loudly. |
| Falcon | A woman speaks and then a kid yells loudly as soft voices speak softly in the background |
| Traditional NLP | A background speaks and then a voices yells loudly as soft woman speak softly in the kid |
| Original | Wood thumping then clanking followed by metal snapping |
| GPT-4 | The sound of metal snapping after wood thumping and clanking |
| LLaMa | Metal snapping is heard, followed by wood thumping and clanking. |
| LLaMa-2 | Metal snapping is heard, followed by wood thumping and clanking. |
| Falcon | The sound of metal snapping after wood thumping and clanking |
| Traditional NLP | thumping Wood then clanking followed by metal snapping |
| Original | A large gun is fired then several beeps goes off and a woman speaks |
| GPT-4 | A large beep is fired then several guns go off and a woman speaks |
| LLaMa | Screaming is heard, followed by a large gun being fired. |
| LLaMa-2 | Screaming is heard, followed by a large gun being fired. |
| Falcon | A large gun is fired then several beeps goes off and a woman speaks |
| Traditional NLP | A several gun is fired then large beeps goes off and a woman speaks |

by the start of an electric toothbrush.', 'An electric toothbrush starts, followed by a man speaking twice, and then clicking sounds.']; If you see, all the modified captions change the temporal order of the 3 events in the original caption. Here is another example: Original Caption: 'A man speaks, followed by a whack and whoosh sound effect, then cheering and shouting from a crowd', Modified Captions: ['A man speaks amidst a whack and whoosh sound effect and cheering and shouting from a crowd', 'A whack and whoosh sound effect is heard, followed by a man speaking, then cheering and shouting from a crowd.', 'A man speaks while a whack and whoosh sound effect is heard, followed by cheering and shouting from a crowd.', 'A whack and whoosh sound effect, along with cheering and shouting from a crowd, is heard before a man speaks.', 'A man speaks and then there is cheering and shouting from a crowd, followed by a whack and whoosh sound effect.']. Similar to the previous example, the temporal ordering of all events in the original caption were changed in the modified captions. For some captions, you will have no modifications, like 'Birds and fowl are heard in the background.' because they have just one event. So in this case just return an empty list. Please don't return captions that translate to the same audio composition. Please only return a json with these lists of 3 modified captions and nothing else. Here is the list of captions:

2. I will provide you with a list. Each item in the list is a caption of an audio file. The caption may have multiple noun-verb combinations, i.e., a noun performing a verb. You need to swap the verbs and associate them with a different noun until the swap and the resulting sentence make sense. For example: you cannot swap an animal sound with a noun describing a human. Here is an example: Original Caption: 'A woman speaks with a baby crying in the background as occasional breathing is heard.' Modified captions: ['A woman crying with a baby speaking in the background as occasional breathing is heard.']. Only one swap was possible with this example. ; Original Caption: 'A baby cries while male singing and music play in the background.' Modified captions: ['A baby sings while

male cries and music play in the background.'] ; Original Caption: 'Human sounds and environmental noise interrupted by occasional taps.' Modified captions: ['Human noise and environmental sounds interrupted by occasional taps.']. If you think more than one swap is possible, return multiple. For some captions, no swap will be possible. This includes captions with just one noun-verb pair or where swaps don't make sense. Original Caption: 'A man speaks, and a dog barks in background noise followed by a shout.' Modified captions: []; Original Caption: 'A cat meows as music plays and a female sings.' Modified captions: []; Original Caption: 'A baby cries.' Modified captions: []; Original Caption: 'A woman talks nearby as water pours.' Modified captions: []. In all these cases, swapping the noun and verbs do not make sense, for example water cant talk and women cant pour. In such cases, return an empty list. Please only return a JSON with these lists for each caption in the list given to you and nothing else. Here is the list of captions:

We generate AudioSet-CompA by prompting GPT-4 for time-aligned compositional captions for audios in the AudioSet strong dataset. The AudioSet strong dataset has time-aligned labels (the start and the end time for each event, e.g.,) for each acoustic event in an audio. Thus, we prompt GPT-4 with these time-aligned labels and ask it to generate a coherent caption from this, and indeed, GPT-4 performs accurately at this task. The prompt used for this task is listed below:

1. I will give you a list of of lists. Each list in the list describes a 10 second audio file in the form of multiple tuples. Each tuple describes a sound event with their starting and ending times in the audio. For example, '(Wind-0.0-10.0)' signifies that Wind was heard through the 10 second audio and '(Tick-6.899-7.01)' signifies a sound of a clock tick was heard from 6.899 to 7.01 span of the 10s audio. The events in the list are temporally aligned. You need to create a one liner caption out of this list of multiple tuples that 1) Follows the temporal sequence of events , 2) The caption in detail describes the compositionality of the audio, i.e. for eg.,, what occurs before what and with what and 3) Make sure you are using grammatical subject-verb-object sentences. Directly describe the sounds and avoid using the word "heard". You should also use your reasoning skills to not blindly follow the temporal events in the list but output a caption of a 10 second audio that explains a plausible real life 10 second event for which you may also ignore some events to achieve. Here are some examples: Input: ['(Laughter-4.881-6.144)', '(Female speech, woman speaking-7.26-8.035)', '(Tick-8.603-8.694)', '(Background noise-0.0-10.0)', '(Speech-1.365-3.569)', '(Breathing-8.948-9.687)']Output: A female laughter is heard followed her talking over a constant background noise in the background. Input: ['(Female speech, woman speaking-2.734-3.783)', '(Music-0.0-8.359)', '(Ocean-4.373-9.399)', '(Rain-0.0-4.086)'], Output: A woman is speaking while while its raininng and music is playing in the background. Input: ['(Female speech, woman speaking-7.354-9.039)', '(Dishes, pots, and pans-9.906-10.0)', '(Mechanisms-0.0-10.0)', '(Generic impact sounds-9.551-9.843)'] Output: Generic mechanism sounds are heard throughout while later on a female starts speaking followed by dishes and pots clanging. Input: ['(Applause-0.0-6.78)', '(Crowd-0.0-10.0)', '(Male speech, man speaking-3.693-4.481)', '(Male speech, man speaking-5.091-5.781)', '(Male speech, man speaking-6.78-7.601)', '(Tick-9.25-9.347)'], Output: A man is heard talking while the crowd is applauding throughout, and following this towards the end a ticking sound is observed. Just return a single list of one liner captions that are (less than 30 words). The list format should be ['caption for audio 1','caption for audio 2',..]. Here is the list of items:

We used the following prompt to generate scenes for modular contrastive learning using GPT-4:

1. I will provide you with a list of unique sound events. I have taken these events from labels in a audio dataset. Generate acoustic scenes that are highly plausible in the real-world. Here is the list of events:

## B.6 MODULAR CONTRASTIVE LEARNING

### B.6.1 TEMPLATE-BASED SYNTHETIC CREATION OF AUDIO-CAPTION PAIRS

Algorithm 1 demonstrates the algorithm for our template-based synthetic audio-caption creation process. For concatenating, we use the $\mathrm{append(.)}$ function in Soundfile library with a crossfade

---

**Algorithm 1** Template Based Audio-Caption Creation

---

**Data:** Acoustic Events Dataset $\mathcal{P} \rightarrow \{\mathcal{A}\,(Audio), \mathcal{L}\,(Label)\}$;

```
// Generate List Of Possible Acoustic Scenes
```
$\mathcal{E} = LLM(Prompt, \mathcal{L})$
```
// Generate Compositional Audio and Fine-grained Positive and Negative
Captions
```
initialize($\mathcal{A}$, $\mathcal{T}^{pos}$, $\mathcal{T}^{neg}$)
**for** $i = 1\,to\,|\mathcal{E}|$ **do**
    initialize($\mathcal{A}^{list}$, $\mathcal{T}_p^{list}$, $\mathcal{T}_n^{list}$)
    **for** $j = 1\,to\,|\mathcal{E}_i|$ **do**
        **if** isAcousticEvent($\mathcal{E}_{i,j}$) **then**
           $(\mathcal{A}^{list}).\,\mathrm{append}(\mathrm{getAudio}(\mathcal{E}_{i,j})), (\mathcal{T}_p^{list}).\,\mathrm{append}(\mathcal{E}_{i,j})$
        **end**
```
        // Generate fine-grained positive and negatives by changing the
         order or operation(+/*)
```
        **else if** isOperation($\mathcal{E}_{i,j}$) **then**
           $a_1,\,a_2 = (\mathcal{A}^{list}).\,\mathrm{pop}(), (\mathcal{A}^{list}).\,\mathrm{pop}()$
           **for** $k = 1\,to\,j - 2$ **do**
               $(\mathcal{T}_p^{list}).\,\mathrm{append}(\mathrm{concatenate}(\mathcal{T}^{list}[k], \mathcal{E}_{i,j}, \mathcal{E}_{i,j-1}))$
               $(\mathcal{T}_n^{list}).\,\mathrm{append}(\mathrm{swapOrderAndChangeOperation}(\mathcal{T}^{list}[k], \mathcal{E}_{i,j}, \mathcal{E}_{i,j-1}))$
           **end**
```
            // Generate Compositional Audio
```
           **if** $(\mathcal{E}_{i,j}).\,\mathrm{isEquals}(``+")$ **then**
               $(\mathcal{A}^{list}).\,\mathrm{append}(\mathrm{concatenateAudio}(a_1, a_2))$
           **end**
           **else if** $(\mathcal{E}_{i,j}).\,\mathrm{isEquals}(``*")$ **then**
               $(\mathcal{A}^{list}).\,\mathrm{append}(\mathrm{overlayAudio}(a_1, a_2))$
           **end**
        **end**
        $(\mathcal{A}).\,\mathrm{append}((\mathcal{A}^{list}).\,\mathrm{pop}())$

```
        // Template based conversion to captions
```
        $(\mathcal{T}^{pos}).\,\mathrm{append}(\mathrm{convertToCaption}(\mathcal{T}_p^{list}))$
        $(\mathcal{T}^{neg}).\,\mathrm{append}(\mathrm{convertToCaption}(\mathcal{T}_n^{list}))$
    **end**
**end**

---

of 10% of the duration of the shorter audio file. For overlay, we employ the algorithm proposed by (Tokozume et al., 2018) to mix two sounds. Additionally, Table 8 shows examples of positives and negatives generated for our modular constrastive learning approach, and Table 9 shows examples of acoustic scenes output by GPT to create the final audio.

Table 8: Template-based fine-grained positive and negative captions for various acoustic scenes,

| Acoustic Scenes | Positives | Negatives |
|---|---|---|
| (Hammer * Jet engine) + Crackle | Hammer and Jet engine and then Crackle
Jet engine and then Crackle
Hammer and Jet engine | Hammer and Crackle and then Jet engine
Hammer and then Jet engine and then Crackle
Jet engine and Crackle
Crackle and then Jet engine
Hammer and then Jet engine |
| (Machine gun + Jackhammer) * Engine knocking | Machine gun and then Jackhammer amidst by Engine knocking
Jackhammer amidst by Engine knocking
Machine gun and then Jackhammer
Machine gun amidst by Engine knocking | Jackhammer and then Machine gun amidst by Engine knocking
Jackhammer and then Engine knocking
Machine gun amidst by Jackhammer
Jackhammer and then Machine gun |
| Whispering * (Female singing + Female speech, woman speaking + Sonic boom) | Whispering overlayed by Female singing succeeded by Female speech, woman speaking succeeded by Sonic boom
Whispering overlayed by Female singing succeeded by Female speech, woman speaking
Whispering overlayed by Female singing succeeded by Sonic boom
Whispering overlayed by Female speech, woman speaking succeeded by Sonic boom
Female singing succeeded by Female speech, woman speaking succeeded by Sonic boom
Female speech, woman speaking succeeded by Sonic boom | Female singing overlayed by Female speech, woman speaking Sonic boom succeeded by Female singing
Whispering succeeded by Female speech, woman speaking succeeded by Sonic boom
Female speech, woman speaking succeeded by Female singing
Female speech, woman speaking succeeded by Female singing
Whispering overlayed by Female singing overlayed by Female speech, woman speaking |

Table 9: Examples of acoustic scenes created for various numbers of audio snippets for final synthetic audio creation in our proposed template-based synthetic audio-caption creation process.

| Number of Acoustic Events | Acoustic Scenes |
| --- | --- |
| 2 | Vehicle * Hammer
Female speech, woman speaking * Tap dance
Crow * Rain
Mechanical bell * Engine starting
Bird * Whimper (dog)
Crow * Roaring cats (lions, tigers)
Doorbell + Engine
Bell + Engine starting
Train whistle + Car
Buzzer + Drawer open or close |
| 3 | Keys jangling * (Howl (wind) + Male singing)
(Tap dance + Applause) * Female speech, woman speaking
Drill * (Computer keyboard + Emergency vehicle)
Chainsaw * (Power windows, electric windows + Train)
(Light engine (high frequency) + Train horn) * Children shouting
(Engine starting + Bell) * Human group actions
(Propeller, airscrew + Siren) * Booing
Train whistle + Dial tone + Children shouting
(Truck + Fusillade) * Howl (wind)
(Cart + Rumble) * Wind noise (microphone) |
| 4 | Coo * (Background noise + Fly, housefly + Speech synthesizer)
Bird vocalization, bird call, bird song * (Stream, river + Roaring cats (lions, tigers) + Speech)
Chicken, rooster * (Environmental noise + Mosquito + Child speech, kid speaking)
(Train + Drill) * (Booing + Wind)
Booing * (Child speech, kid speaking + Male singing + Environmental noise) |

## B.7 HYPER-PARAMETER SETTINGS

**Vanilla CLAP Training.** For vanilla contrastive pre-training with CompA-661k, we use a batch size of 24, and Adam optimizer with a learning rate of 1e-4, and warm-up of 3200 steps, and train for 45 epochs. For training with compositionally aware hard negatives, we start with vanilla CLAP weights and train for 20 epochs with no warm-up. We follow a similar setup for modular contrastive training.

## B.8 BASELINES

For retrieval-based evaluation (text-to-audio and audio-to-text), we compare CompA-CLAP with six baselines:

**MMT** (Oncescu et al., 2021) introduces the task of retrieving audio using free-form natural language queries. The authors argue that this is a more intuitive and flexible way to search for audio than traditional methods, which rely on text annotations, and also demonstrate the benefits of pre-training on diverse audio tasks.

**ML-ACT** (Mei et al., 2022) studies the impact of different metric learning objectives on the audio-text retrieval task and finds that NT-Xent loss is a promising approach that achieves stable performance across different datasets and training settings, and outper-forms the popular triplet-based losses. Metric learning objectives are commonly used to train cross-modal retrieval models by mapping data to an embedding space where similar data are close together and dissimilar data are far apart.

**CLAP** (Deshmukh et al., 2022a) introduces a framework for audio retrieval that uses a contrastive learning objective and two audio encoders to connect language and audio content.

**CLAP-LAION** (Wu* et al., 2023) proposes a pipeline for contrastive language-audio pre-training to develop an audio representation by combining audio data with natural language descriptions. They construct a contrastive language-audio pre-training model by considering different audio encoders and text encoders. They incorporate the feature fusion mechanism and keyword-to-caption augmentation into the model design to further enable the model to process audio inputs of variable lengths and enhance performance.

Table 10: Performance comparison of CompA-CLAP with baselines on Real-world / Synthetic distribution in CompA-attribute benchmark.

| | CompA-attribute | | |
|---|---|---|---|
| **Model** | **Text** | **Audio** | **Group** |
| Human | 80.30 / 84.10 | 82.40 / 86.20 | 79.80 / 83.50 |
| Random | 25.0 / 21.0 | 25.0 / 22.0 | 16.67 / 15.33 |
| CLAP | 33.27 / 37.21 | 6.14 / 8.12 | 4.66 / 6.23 |
| CLAP-LAION | 34.78 / 38.43 | 6.52 / 8.94 | 5.07 / 7.82 |
| CompA-CLAP *(ours)* | $\mathbf{44.28}_{\pm 0.02}$ / $\mathbf{46.34}_{\pm 0.03}$ | $\mathbf{22.52}_{\pm 0.04}$ / $\mathbf{24.31}_{\pm 0.06}$ | $\mathbf{15.13}_{\pm 0.02}$ / $\mathbf{17.10}_{\pm 0.04}$ |

**Wav2CLIP** (Wu et al., 2022) proposes a pre-trained audio representation learning method that distills knowledge from Contrastive Language-Image Pre-training (CLIP). Wav2CLIP projects audio into a shared embedding space with images and text, which enables multimodal applications such as zero-shot classification and cross-modal retrieval.

**AudioClip** (Wu et al., 2022) is an extension of the CLIP model that can handle audio in addition to text and images by incorporating the ESResNeXt audio-model in the CLIP framework. It was trained on the AudioSet dataset, which contains millions of audio clips with corresponding labels.

# C ADDITIONAL RESULTS

## C.1 SELECTING $K$

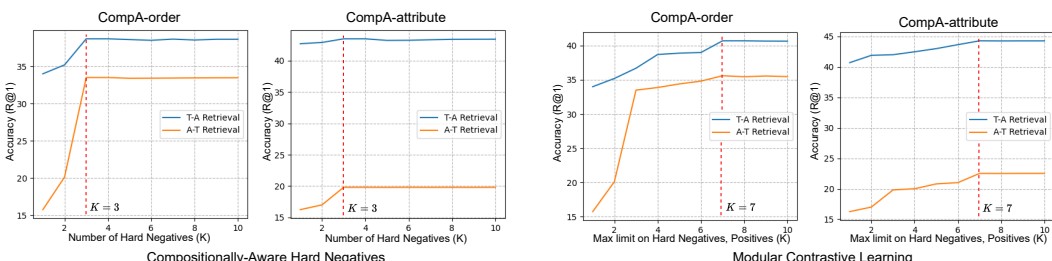

Figure 8: Ablation on varying $K$ on CompA-order/attribute benchmark

Figure 8 provides insight regarding how to select $K$ which is the number of hard negatives for Compositional-Aware Hard Negative training, and also the max limit for selecting generated fine-grained positives as well as negatives for Modular Contrastive Learning.

## C.2 MEASURING QUALITY OF NEGATIVE SAMPLES

In this section, we measure the quality of negative samples generated in AudioSet-CompA. For each caption $t_i \in \mathcal{T}$, we randomly sample 1000 captions $t_i^{neg}$ from the same dataset, which act as negatives. To compare the quality of the generated negatives $t_i^{hard}$ vs $t_i^{neg}$, we encode these captions with sentence transformer (Reimers, 2023) and calculate relative difference between the cosine similarity of $sim(t_i, t_i^{hard})$ and $sim(t_i, t_i^{neg})$ which comes out to be 70.7%. From this, we conclude that the overall quality of generated hard negatives is superior to the negatives sampled from the dataset.

## C.3 COMPARING SYNTHETIC VS ORIGINAL COUNTERPARTS DURING EVALUATION

In this section, we compare the performance of CompA-CLAP on the original subset Vs the synthetic subset of the CompA-Attribute benchmark. While the original subset, also known as real-world distribution, is derived from the AudioSet (Gemmeke et al., 2017), the *synthetic distribution* is prepared using WavJourney (Liu et al., 2023d) and constitutes about 25% of the CompA-Attribute benchmark. Table 10 shows the result across both the distribution. Compared to the baselines,

Table 11: Performance CompA-CLAP across various acoustic scenes in CompA-Order benchmark.

| Acoustic Scenes | CompA-order | | |
| --- | --- | --- | --- |
| | Text | Audio | Group |
| Human Speech, Land Animal Sounds | 71.22 | 68.41 | 63.80 |
| Human Speech, Bell Sounds | 77.11 | 72.53 | 68.12 |
| Human Speech, Human Sound | 21.23 | 11.41 | 10.13 |
| Tools and Equipment Sound, Construction Sound | 20.12 | 10.42 | 8.10 |

Table 12: Analyzing CompA-CLAP Performance: Exploring variations across selected acoustic events based on it's frequency in AudioSet (training set)

| Acoustic Scenes | CompA-order | | |
| --- | --- | --- | --- |
| | Text | Audio | Group |
| Male speech, Generic Impact Sound | 61.32 | 58.42 | 52.13 |
| Woman speaking, Music | 65.21 | 60.14 | 58.23 |
| Video game sound, Plop | 16.23 | 8.21 | 6.24 |
| Eruption, Stream | 12.32 | 7.13 | 5.21 |

we observe a consistent boost in the performance across both distributions in the CompA-attribute benchmark.

## C.4 ERROR ANALYSIS

- The analysis, shown in Table 11 evaluates CompA-CLAP performance across combinations of acoustic events, considering acoustic similarities between audios. Notably, the model excels in scenes with acoustically dissimilar audios (e.g., Human Speech and Land Animal Sounds), but its performance dips in scenes with similar audio types (e.g., Human Speech and Human Sound).

- Table 12 also highlights the long-tailed problem in AudioSet, where the CompA-CLAP underperforms in audio with acoustic events seen less during training, for instance Video game sound, Stream, Plop, Eruption, etc.

- Table 13 illustrates the qualitative analysis, showcasing instances of significant improvement in CompA-CLAP compared to other baseline models within the acoustic compositions of the CompA-order and CompA-attribute benchmarks.

Table 13: Some examples of acoustic compositions in the CompA-order and CompA-attribute benchmarks where we observed significant improvement in CompA-CLAP compared to the other baselines

| CompA-order (Acoustic Scene) | CompA-attribute (Acoustic Scene) |
| --- | --- |
| Frog croaking followed by music playing. | A man speaks then a woman panics |
| Man speaking before bell ringing. | A baby laughs while a woman make sounds |
| A man speaking followed by gunshot | A child sneezes and an adult laughs |
| Tiger growling followed by people speaking. | Adult female is speaking and a young child is crying |

Table 14: Examples of negatives generated for contrastive training with compositionally-aware hard negatives. Sentences marked with gray show negatives where the order of occurrence of the acoustic events in the audio differs, and sentences marked with orange show negatives where the attribute between multiple acoustic events is switched.

| Label | Caption |
|---|---|
| Original Caption: | Men speaking then sneezing. |
| Negative: | Men sneezing then speaking. |
| Original Caption: | A child screams, an adult male is talking and vehicles are revving. |
| Negative: | Vehicles are revving preceding a child screams and an adult male is talking. |
| Negative: | An adult male talks preceding vehicles rev and a child screams. |
| Negative: | A child talks, an adult male is screaming and vehicles are revving. |
| Original Caption: | A crowd is cheering and shouting, thumping occurs, an adult female speaks, and an adult male speaks. |
| Negative: | An adult female speaks following the cheering and shouting crowd, followed by thumping and an adult male speaking |
| Negative: | Thumping occurs, an adult male speaks, then a crowd is heard cheering and shouting and an adult female speaks. |
| Negative: | A crowd is speaking and shouting, thumping occurs, an adult female cheers, and an adult male speaks |
| Original Caption: | High pitched drilling followed by rustling. |
| Negative: | Rustling sounds followed by high pitched drilling |
| Negative: | Rustling and high pitched drilling happening together |
| Negative: | High pitched rustling followed by drilling |
| Original Caption: | The entire time a loud static sound is joined with a constant clicking |
| Negative: | The entire time a constant clicking accompanies a loud static sound |
| Negative: | Simultaneously, a loud static sound and a constant clicking are heard |
| Negative: | The entire time a loud clicking is joined with a constant static sound |

```python
def build_mask(sim_shape, no_positive_list, no_negative_list):

    """
    Build Mask for generated positive and negative Captions

    Args:
        sim_shape (List): shape of Similarity matrix
        no_positive_list (torch.Tensor): Number of positive captions per
            audio.
        no_negative_list (torch.Tensor): Number of negative captions per
            audio.

    Returns:
        mask_tensor (torch.Tensor)
    """
    #create index_map and index_range to identify
    #the specific ranges for each texts wrt generated cations (pos/neg)
    index_map = []
    no_gen_caption = torch.cat(no_positive_list, no_negative_list)

    for i in range(len(no_gen_caption)-1):
        if not index_map:
            index_map.append(no_gen_caption[i]+1)
        else:
            index_map.append(index_map[i] + no_gen_caption[i] + 1)
    index_range = [[x,x+y] for x,y in zip(index_map, no_gen_caption)]

    #intialize the mask
    mask_tensor = float('-inf') * torch.ones(sim_shape)
    caption_size = sim_shape[0]

    for i in range(caption_size):
        gen_index = 0 #Get index w.r.t the generated pos/neg cations
        batch_index = 0 #Get index w.r.t the original captions

        for k, (start_index, end_index) in enumerate(index_range):
```

```
35              if start_index <= k and k <= end_index:
36                  gen_index = k
37                  batch_index = start_index
38                  break
39
40          if i == batch_index:
41              mask_tensor[i,:] = 1
42          else:
43              mask_tensor[i][gen_index] = 1
44
45      return mask_tensor
```

Listing 1: Source Code To Build Mask

