# OpenReview forum: "CompA: Addressing the Gap in Compositional Reasoning in Audio-Language Models"
_ICLR.cc/2024/Conference — ICLR 2024 poster_

### Official Review · Reviewer_Cw3v · 2023-10-28

**Soundness:** 3 good
**Presentation:** 3 good
**Contribution:** 3 good
**Rating:** 6
**Confidence:** 4

**Summary:**

The authors propose and curate a new benchmark for evaluating the compositional reasoning capability in audio-language models. And further show that to train with curated data and modular contrastive with hard positives and negatives help to improve the compositional reasoning capability.

**Strengths:**

- The curation of CompA-order and CompA-attribute benchmarks are relevant and needed contributions to the community. This helps to provide another perspective to examine the properties and limitations of current SOTA methods for audio-language models.
- To provide both quantitative analysis of the current limitations in 2.1 and qualitative analysis in the last paragraph of 3.1 both provide solid motivations of this work.

**Weaknesses:**

- It is a bit difficult to get an entire picture of number of different datasets curated and how each of them are curated, a lot of the information are buried in the appendix. Consider in the main paper including a table describing different datasets curated, how many of them for each subset used for training, and evaluation, their source, and methods (leverage ChatGPT, replace the sounding object, swap the orders, etc.), and then refer to the table in the narratives.
- The data curation is one of the main contribution for this work, however currently a lot of details are left in the appendix. Consider integrating more information such as the source datasets, and some manipulated examples to the main narratives, this can help the reader to understand this work early on.
- A minor suggestion: The figures and tables are far from the actual text in the appendix, consider reorganize them to make it easier to read.
- A minor comment: A lot of the texts in the tables and the figures are very small and difficult to read.

**Questions:**

- Regarding the data curated for training, there are CompA-661k, and AudioSet-compA, another 110k audio-text pairs for complex compositional audios? How are these different subsets used in different training scenario? It might worth adding a table to show clearly what combination of datasets, training strategies, whether there is hard negative and modular contrastive utilized, and link them directly to your abbreviation in the result table (CLAP, CompA-CLAP, CLAP-CompA-661k, etc.) Currently it is not easy to associate from the narratives to exactly how each variation in the result tables comes from.
- In Table 1, what is the difference between CLAP (ours) and CLAP-CompA-661k (ours)?
- For the common mistakes mentioned in the discussion, is it possible to perform some error analysis? Or confusion matrices to show these more quantitatively?

---

> ### Author Response · Authors · 2023-11-13
> **Response to Reviewer Cw3v (1/2)**
>
> We thank you for your thorough review and constructive feedback. We have tried to address each of your concerns point by point.
>
> ## Weaknesses:
> >Q. It is a bit difficult to get an entire picture of number of different datasets curated and how each of them are curated, a lot of the information are buried in the appendix. Consider in the main paper including a table describing different datasets curated, how many of them for each subset used for training, and evaluation, their source, and methods (leverage ChatGPT, replace the sounding object, swap the orders, etc.), and then refer to the table in the narratives.
>
> **Ans:** Thank You for your suggestion. Due to space limitations, despite our best efforts, we could not include such a table in our revised version due to space constraints. However, we have made the following changes to make it clearer:
> - We have added Table 7 in the Appendix of our paper. This Table describes all the data used in our experiments, including their source and the stage of training in which they were used. We also show the Table in the Questions section.
> - We have made several writing changes in all sections to make it clearer which data is being used in which stage of training. The changes can be seen in the revised version of our paper.
>
> >Q. The data curation is one of the main contributions of this work. However, currently, a lot of details are left in the appendix. Consider integrating more information, such as the source datasets and some manipulated examples to the main narratives, this can help the reader to understand this work early on.
>
> **Ans:** Thank You for your suggestion. We have made the following changes to incorporate more details about data curation both in the main paper and appendix:
> 1. We expanded the details about the annotation in Section 2 and included more details about the exact annotation process.
> 2. In Appendix B3, we provide more details about annotator demographics.
> 3. In Appendix B3, we list some rules adhered to during the annotation process.
>
> We would also like to point out that both our proposed contrastive learning approaches are novel and also serve as the main contributions of the paper. Compa-CLAP serves as the first step toward solving compositional reasoning in audio-language models. Thus, to accommodate everything within the 9-page limit, we made a division by allocating 3 pages to the dataset description and 4 to Compa-CLAP.
>
> >Q. A minor suggestion: The figures and tables are far from the actual text in the appendix, consider reorganize them to make it easier to read.
>
> **Ans:** Thank You for your suggestion. We have reorganized the Figures and Tables in the appendix of the revised version of our paper to make it easier to read.
>
> >Q. A minor comment: A lot of the texts in the tables and the figures are very small and difficult to read.
>
> **Ans:** Thank You for your suggestion. We have revised the text in our figures in the revised version of our paper to make them clearer.

---

> ### Author Response · Authors · 2023-11-13
> **Response to Reviewer Cw3v (2/2)**
>
> ### Questions:
>
> >Q. Regarding the data curated for training, there are CompA-661k, and AudioSet-compA, another 110k audio-text pairs for complex compositional audios? How are these different subsets used in different training scenario? It might worth adding a table to show clearly what combination of datasets, training strategies, whether there is hard negative and modular contrastive utilized, and link them directly to your abbreviation in the result table (CLAP, CompA-CLAP, CLAP-CompA-661k, etc.) Currently it is not easy to associate from the narratives to exactly how each variation in the result tables comes from.
>
> We apologize for the confusion. Here is a table to make things clearer. We have also added this table to our Appendix as Table 7.
>
> | Dataset                     | Source                                  | Experiment                                                                       |
> |--|---|---|
> | Pre-training                |                                         |                                                                                  |
> | CompA-661K                  | Multiple (refer to Table 6 in Appendix) | Vanilla Contrastive Pre-training (Section 3.2)                                   |
> | CompA-AudioSet              | AudioSet Strong [1]                     | Contrastive Pre-training with compositionally aware hard negatives (Section 3.3) |
> | Modular Contrastive Dataset | AudioSet Strong [1]  (but synthetic)                   | Modular Contrastive Learning (Section 3.4)                                       |
> | Evaluation                  |                                         |                                                                                  |
> | CompA-order                 | AudioSet Strong [1]                           | Evaluation Dataset                                                               |
> | CompA-Attribute             | AudioSet Strong [1]                            | Evaluation Dataset                                                               |
> | CompA-Attribute             | WavJourney [3]                          | Evaluation Dataset                                                               |
>
> Table 1: List of various datasets used in our experiments, their sources, and the specific experiments they are used in.
>
> Note: As also mentioned in the paper, no audio used to create our CompA benchmark was included in the training set at any stage. For modular contrastive learning, we only take individual sliced acoustic events from the AudioSet Strong (not the entire audio), to generate our own synthetic data.
>
> >Q. In Table 1, what is the difference between CLAP (ours) and CLAP-CompA-661k (ours)?
>
> **Ans:** We apologize for the mistake and thank you for pointing this out. CLAP (ours) and CLAP-CompA-661k (ours) are the same models, and CLAP-CompA-661k (ours) in Table 1 (right) is a typo. Since we trained our version of CLAP on CompA-661k, we had initially named it CLAP-CompA-661k but later changed it to CLAP (ours).
>
> >Q. For the common mistakes mentioned in the discussion, is it possible to perform some error analysis? Or confusion matrices to show these more quantitatively?
>
> **Ans:** Thank You for the suggestion. We have added both quantitative and qualitative comparisons in Appendix C.4 in Tables 11, 12 and 13. In terms of qualitative analysis, we have highlighted specific acoustic compositions in the CompA-order and CompA-attribute benchmarks where we observed significant improvement in CompA-CLAP compared to the other baselines.
>
> | CompA-order (Acoustic Scene)                | CompA-attribute (Acoustic Scene)                     |  |
> |---|---|----|
> | Frog croaking followed by music playing.    | A man speaks then a woman panics                     | |
> | Man speaking before bell ringing.           | A baby laughs while a woman make sounds              | |
> | A man speaking followed by gunshot          | A child sneezes and an adult laughs                  | |
> | Tiger growling followed by people speaking. | Adult female is speaking and a young child is crying | |
>
> Table 2: Some examples of acoustic compositions in the CompA-order and CompA-attribute benchmarks where we observed significant improvement in CompA-CLAP compared to the other baselines
>
> **References:**
>
> [1] Hershey, Shawn, et al. "The benefit of temporally-strong labels in audio event classification." ICASSP 2021-2021 IEEE International Conference on Acoustics, Speech and Signal Processing (ICASSP). IEEE, 2021.
>
> [2] Gemmeke, Jort F., et al. "Audio set: An ontology and human-labeled dataset for audio events." 2017 IEEE international conference on acoustics, speech and signal processing (ICASSP). IEEE, 2017.
>
> [3] Liu, Xubo, et al. "Wavjourney: Compositional audio creation with large language models." arXiv preprint arXiv:2307.14335 (2023).

---

> ### Author Response · Authors · 2023-11-16
> **Request to review the rebuttal**
>
> Dear reviewer Cw3v,
>
> Thank you for the time you spent reviewing our paper. We have submitted our response to your concerns, including a revised version of our paper. Please let us know your comments and if you have any more concerns. Thank You again!

---

> > ### Author Response · Authors · 2023-11-18
> > **Request to review the rebuttal**
> >
> > Dear reviewer Cw3v,
> >
> > Thank you for the time you spent reviewing our paper. We have submitted our response to your concerns, including a revised version of our paper incorporating all changes requested. Please let us know your comments and if you have any more concerns. Thank You again!

---

> > > ### Comment · Reviewer_Cw3v · 2023-11-18
> > >
> > > Thank the authors for your efforts to address the questions and prepare for open sourcing the benchmark. I am increasing my ratings.

---

> > > > ### Author Response · Authors · 2023-11-18
> > > > **Thank You!**
> > > >
> > > > We are glad we could address your concerns. We thank you again for the detailed review and insightful comments, which helped us improve the quality of the paper.

---

### Official Review · Reviewer_Xast · 2023-10-30

**Soundness:** 3 good
**Presentation:** 3 good
**Contribution:** 3 good
**Rating:** 6
**Confidence:** 4

**Summary:**

Summary: This paper presents a new benchmark, CompA, for measuring two forms of compositionality in audio-language models. These are ordering (sequencing of acoustic events within a clip) and attribute binding (determining which objects perform acoustic sources make which sounds in a clip). They also propose a novel pipeline to train a new contrastive model, CompA-CLAP, by augmenting existing datasets with (a) hard negatives in terms of both forms of compositionality (ordering and attribute binding) and (b) data augmentation by concatenating and overlaying audio clips with known labels, where all text is generated via LLM. Using their proposed benchmark, they show improvements from the novel data pipeline.

**Strengths:**

I am posting my full review here, as I believe it easier to discuss both strengths and weaknesses together.

# Overall

Overall, the paper is an ambitious effort to propose a new benchmark measuring a known limitation of existing audio-language models, and also to achieve improved performance on this benchmark. However, reading the paper, it also feels that important details are sometimes lacking both on the benchmarking and the modeling/empirical sides of the paper, in part due to this decision to cover so much ground. It does not appear that the missing information is available in the supplementary material either. The proposed method also performs below random baseline on 2 of the 6 proposed benchmark metrics, but this does not appear to be discussed in the main text. I recommend adding some important clarifying details about the benchmark, clarifying experimental decisions and details, and reducing the "sales pitch" framing of the paper. I am enthusiastic about the direction and would encourage the authors to continue to refine the benchmark and their work. However, I feel that major revisions to the paper, and perhaps the benchmark, are currently in order, and that it is probably not ready for acceptance in its current form.

# Major Comments

* The tone of the paper often leans too strongly toward "sales pitch" for my taste. I would recommend to tone down this language, and focus on making quantifiable and verifiable claims based on the data and methods in the paper. Just a sampling of the phrases the authors use to dismiss other work, minimize problems, or promote theirs:

  - "only a word changed" (a few sentences after the claim "the audios have the exact same words")
  - "it is trivial for ALMs to perform well on these benchmarks"
  - "CLAP undergoes only minor degradation in retrieval performance"
  - "25% synthetic...but highly curated by expers" (what is *highly* curated?)
  - "a trivial but ineffective solution"
  - "a highly custom prompt"
  - "Our version outperforms ... by significant margins" (no measures of significance used)

* Since the authors are not only introducing a novel model, but actually proposing their dataset to serve as a benchmark, more justification and detail are needed. For example:
  - There is no mention in the paper of where the benchmark can be accessed, how it can be used by other works, what format it is provided in, etc. This is essential information for a public benchmark (it is ok to maintain anonymity by describing where the benchmark *will* be available upon acceptance, if the description is sufficiently clear).
  - The two components of the benchmark consist of only 400 and 200 samples respectively. Is this sufficiently large for a reliable benchmark for the field? Even existing benchmarks in the audio-text domain consist of thousands of examples (AudioSet, AudioCaps, MusicCaps). If 200-400 samples is considered sufficient, please provide empirical evidence. For reference, a Clopper-Pearson confidence interval for even 400 samples would be quite wide: a width of roughly 0.1 for a 95% CI depending on the approximation used, when the success rate is 50%.
  - It is critical that the paper provide more detail on the annotation process in the main text. The current version provide almost no information, except that four annotators participated. Please describe, for example: number of annotators *per example*; the exact annotation procedure performed. Rater agreement metrics (in supplementary) would be useful but are not required. More detail is also needd regarding the screenshot in Figure 6; it is not clear at all what task the raters are even performing. Much of Section A.3 is in the passive tense, so it is also hard to tell who performed what tasks ("a manual inspection was done","Wavjourney was employed"). Many of these details should also appear in the main text.
  - The abstract states that the benchmark consists of "a majority of real-world samples", and elsewhere it is mentioned that 90% of audio are from real-world samples. (1) Can you provide results separated by real-world vs. not real-world? (2) What is the nature of the "non real-world" data?
  - The choice of metrics seems not entirely clear. It would be helpful to either (1) cite works which already use similar metrics, to motivate their used based on wide preexisting adoption in the ALM community, or (2) provide more detailed motivation for why the metrics are appropriate. For example, why is "group score" the conjunction of audio score and text score and not, e.g., their harmonic mean in the style of an F-score?
  - There are no quantifiable measurements of incertainty or variability in the comparisons presented in the paper. The authors say that their results are averaged over 3 random seeds; can they provide the (variance, stddev) over these seeds? What about Clopper-Pearson or other appropriate confidence intervals for the metrics reported? As is, it is impossible to assess the significance of the differences between metrics in Table 2.

* Some important experimental design decisions in the paper are not clearly motivated and not validated with empirical studies. This includes:
  - Why did the authors only fine-tune CLAP (when they have already shown the CLAP weights to be ineffective) instead of training it from scratch, when the authors apparently have access to the full original pretraining datasets?
  - Why do the authors fine-tune two separate times, instead of performing a single phase of joint fine-tuning?
  - Why does the fine-tuning occur (hard negatives (3.3) --> synthetic pairs (3.4)) instead of the reverse ordering?

All of these decisions could be validated empirically, but I do not see these results in the paper.

* Both examples in Figure 4 both seem to be flawed training example. (1- Left) The negatives both appear to be correct. All of these examples (the true caption, and the negatives) describe three sound events occurring simultaneously. (2 - right) "Tiger roars amidst thunker" and "Tiger roar followed by" cannot both be positive labels for the same sample - only one can be correct. Both of these issues seem to raise concerns about the quality of the data used here; particularly since the "highly custom prompt" described does not appear to be shared in the paper.

* Having the "CompA-CLAP" row in Table 2 be bold is misleading. In particular: "CLAP (ours)" outperforms ComA-CPA in attribute audio score. Additionally, CompA-CLAP performs below random baseline on audio and group scores in CompA-attribute. This should be highlighted clearly in the paper and abstract (which currently says that "CompA-CLAP significantly improves over all our baseline models on the CompA benchmark" but does not mention its below-chance performance on 2 of the 6 benchmark task metrics).

* The paper makes several additional changes to the CLAP model

# Minor Comments


* "Sequence" and "attribute binding" should be defined clearly and early in the paper (ideally before these phrases are used).

* Caption in Figure 2 should describe the metrics being presented in the Figure clearly. The Figure is also missing clear y-axis labels. The phrase "minor" is not quantifiable. This figure could also be improved by (a) measures of variability such as error bars or Clopper-Pearson confidence intervals or (b) baselines to show that the CLAP models actually degrade to "trivial" performance (is R@1 of 0.42 on AudioCaps val, or R@1 of 0.35 on AudioCaps test, "trivial"?).

* A lot could be done to make the paper more self-contained. As-is, it is missing a great deal of relevant background information, which is exacerbated by the deferral of the related work to Section 5.

* "to assure high quality, we don’t concatenate or overlay random acoustic events but ask an LLM to create unique audio scenes based on the available labels" - please clarify both the overall procedure here, and how this assures high quality.

* "where augmenting LAION-audio-630K with 2 million audios from AudioSet improved performance on benchmarks only marginally." Please clarify what is meant in this sentence.

* The "experimental protocol" sections in 3.2 and 2.4 here do not describe protocols; please clarify or remove these sections. I would recommend a single "experimental protocol" section, since these subsections only describe individual components of the overall experimental setup.

# Typos etc.

* Section 1 should reference that prior work is deferred to, and discussed in, Section 5.
* "attribute-binding" vs "attribute binding" both used in the paper.
* "The group score evaluates of the model performed well for the evaluation instance in the benchmark" - this is not a meaningful description.
* 3.4: "due the the inherenly complex and non-linear nature of audios in the wild" - please clarify what is meant here, and how nonlinearity relates to the hardness of an example.

# Edit after discussion

I have increased my scores for soundness (2->3) and presentation (2->3) along with my overall rating (3->6) after the author responses, which made significant changes to the paper which both improved it and addressed several of my concerns.

**Weaknesses:**

See above.

**Questions:**

See above.

---

> ### Author Response · Authors · 2023-11-14
> **Response to Reviewer Xast (1/6)**
>
> We thank you for your thorough review and constructive feedback. We have tried to address each of your concerns point by point.
>
> ### Major Comments:
> >Q. The tone of the paper often leans too strongly toward "sales pitch" for my taste. I would recommend to tone down this language, and focus on making quantifiable and verifiable claims based on the data and methods in the paper. Just a sampling of the phrases the authors use to dismiss other work, minimize problems, or promote theirs:
>
> >- "only a word changed" (a few sentences after the claim "the audios have the exact same words")
>
> **Ans:**  The two claims refer to two different types of captions.  The captions for the audios with two pairs have the exact same words but in a different order. For audios with three pairs, only a single word that defines the preposition between the events is changed. We have incorporated this change in the revised version of the paper.
>
> >- "it is trivial for ALMs to perform well on these benchmarks"
>
> **Ans:** We have replaced this instance of the word “trivial” with 'easy' to convey the intended meaning. We have made the necessary changes in the revised version of our paper.
>
> >- "CLAP undergoes only minor degradation in retrieval performance"
>
> **Ans:**  We have changed the wording of the caption and also added metrics to it in the revised version of our paper (Figure 1).
>
> >-"25% synthetic...but highly curated by expers" (what is highly curated?)
>
> **Ans:** The term "highly curated" implied a strong level of involvement and careful selection by experts in the generation process and validation process. We have addressed this by using clearer language, specifying that the audios are "carefully validated by experts who also supervise the generation process." This revision provides a more detailed and precise description of the expert's involvement in both the validation and supervision of the synthetic audio generation.
>
> >- "a trivial but ineffective solution"
>
> **Ans:**  We have replaced this instance of the word ‘trivial’ with the word “simple” to convey the intended meaning. We have made the necessary changes in the revised version of our paper.
>
> >- "a highly custom prompt"
>
>  **Ans:**  The term "highly" was used to emphasize the thorough and thoughtful nature of the prompt customization process. However, to provide a clearer and more concise description and tone down the language, we have now simplified it to "custom prompt," which still conveys the idea that the prompt was carefully tailored for the task without explicitly using the word "highly." (Section 2.3)
>
> >- "Our version outperforms ... by significant margins" (no measures of significance used)
>
> **Ans:**  Our version of CLAP outperforms [1] on all existing retrieval benchmarks for both text-to-audio and audio-to-text retrieval by 0.15%-4.67%, and CompA-order and CompA-attribute by 11.85%-23.8%. We have incorporated this change in the revised version of the paper.
>
> >Q. Since the authors are not only introducing a novel model, but actually proposing their dataset to serve as a benchmark, more justification and detail are needed. For example:
>
>  >- There is no mention in the paper of where the benchmark can be accessed, how it can be used by other works, what format it is provided in, etc. This is essential information for a public benchmark (it is ok to maintain anonymity by describing where the benchmark will be available upon acceptance, if the description is sufficiently clear).
>
> **Ans:** Post acceptance, we plan to release the entire dataset on our GitHub with CC-BY-NC 4.0 License. Together, with the benchmark, we also plan to release all evaluation codes and training codes for benchmark.
>
> To ensure uniformity in scores, we plan to release a website like the SUPERB Benchmark for spoken language processing tasks (https://superbbenchmark.org/). Researchers will be able upload their models for automatic evaluation and see how their models performs on private and public leaderboards. Since evaluation can be done on a CPU, this would not be a great overhead. The leaderboard will host scores by averaging across 3 runs and report standard deviation across scores.
>
> Note: We are not authors of the SUPERB benchmark.
>
> For now, we have provided a sample of our dataset in a presentation (.pptx file) in the supplementary materials and have also included an anonymous link to access the complete dataset in the supplementary readme file.
>
> **References:**
>
> [1] Yusong Wu, Ke Chen, Tianyu Zhang, Yuchen Hui, Taylor Berg-Kirkpatrick, and Shlomo Dubnov. Large-scale contrastive language-audio pretraining with feature fusion and keyword-to-caption augmentation. In ICASSP 2023-2023 IEEE International Conference on Acoustics, Speech and Signal Processing (ICASSP), pp. 1–5. IEEE, 2023.

---

> > ### Author Response · Authors · 2023-11-14
> > **Response to Reviewer Xast (2/6)**
> >
> > >Q. The two components of the benchmark consist of only 400 and 200 samples respectively. Is this sufficiently large for a reliable benchmark for the field? Even existing benchmarks in the audio-text domain consist of thousands of examples (AudioSet, AudioCaps, MusicCaps). If 200-400 samples is considered sufficient, please provide empirical evidence. For reference, a Clopper-Pearson confidence interval for even 400 samples would be quite wide: a width of roughly 0.1 for a 95% CI depending on the approximation used, when the success rate is 50%.
> >
> > **Ans:** Thank You for the important question. As also mentioned in the first 2 paragraphs of Section 2 of our paper, our benchmark is inspired by the Winoground dataset proposed by Thrush et al. (2022), a pioneering work in visio-linguistic compositional understanding is the Winoground dataset. The benchmark proposed by Thrush et al. (2022) has 400 instances, where each instance has 2 image-caption pairs. Our dataset has 200 more samples than theirs.
> >
> > A note on the difficulty of annotation for the CompA benchmark. We also acknowledge that real-world audio samples that satisfy the needs of our benchmark are much more difficult to obtain than images. CV also benefits from large-scale, openly available datasets from where a benchmark can be extracted. This is not the same for audio. Additionally, synthetic image generation pipelines are also much superior in quality and composition understanding than synthetic audio generation pipelines, which motivates us not to use them.
> > In designing our study on understanding compositionality in audio, we carefully considered various factors, including the complexity of the task, the diversity of compositional variations, and the statistical reliability of our findings.
> > While we acknowledge that a dataset size of 600 (400+200) samples may seem relatively smaller in certain contexts, we want to highlight the following considerations that influenced our decision:
> >
> > - Task Complexity: The nature of understanding compositionality in audio is intricate, involving various aspects and potential nuances. We aimed to cover a broad spectrum of compositional variations, and a larger dataset size of 600 samples allows us to capture a more comprehensive range of scenarios.
> > - Statistical Reliability: Recognizing the impact of dataset size on statistical reliability, we aimed to strike a balance between the desire for a larger dataset and the need for meticulous data annotation and validation. The dataset size was chosen to provide meaningful insights while ensuring the quality of each annotated sample.
> > - Resource Constraints: Practical constraints, such as time, cost, and available resources, also influenced our decision on the dataset size. We wanted to ensure a feasible and thorough annotation process within the given constraints.
> >
> > We are conscious of the trade-offs involved and have taken steps to mitigate potential limitations. We believe that the dataset size, in conjunction with the careful consideration of task complexity and statistical reliability, provides a valuable foundation for our study. As also mentioned in our Conclusion and Future work section, we aim to expand the size of our dataset in the future.
> >
> > >Q. It is critical that the paper provide more detail on the annotation process in the main text. The current version provide almost no information, except that four annotators participated. Please describe, for example: number of annotators per example; the exact annotation procedure performed. Rater agreement metrics (in supplementary) would be useful but are not required. More detail is also needd regarding the screenshot in Figure 6; it is not clear at all what task the raters are even performing. Much of Section A.3 is in the passive tense, so it is also hard to tell who performed what tasks ("a manual inspection was done","Wavjourney was employed"). Many of these details should also appear in the main text.
> >
> > **Ans:** We thank you for the suggestion. We have added more necessary details in the main text for dataset annotation in the revised version of our paper. We have also expanded the Appendix with additional details like:
> > - We expand details about the annotation in Section 2 and include more details about the exact annotation process.
> > - In Appendix B3, we provide more details about annotator demographics.
> > - In Appendix B3, we list down some rules adhered to during the annotation process.
> >
> > We list down annotation rules, illustrate the snapshot of the annotation tool, and provide more information on the annotator backgrounds in Appendix B.3.

---

> > > ### Author Response · Authors · 2023-11-14
> > > **Response to Reviewer Xast (3/6)**
> > >
> > > >Q. The abstract states that the benchmark consists of "a majority of real-world samples", and elsewhere it is mentioned that 90% of audio are from real-world samples. (1) Can you provide results separated by real-world vs. not real-world? (2) What is the nature of the "non real-world" data?
> > >
> > > **Ans:** By "non real-world" we mean synthetically generated audio. This audio is generated using WavJouney. WavJourney expects customized prompts for audio generation, and these prompts were written by the 4 annotators (more detail in Paragraph 1 of Section Y and Appendix C.2). As also mentioned in Section 2.2 and Section 2.3, CompA-order has all-natural audios and only 25% of the audios in CompA-attribute are synthetically generated. This was done due to the lack of real-world audio in AudioSet, which met our requirements.
> > >
> > > |                   | CompA-Attribute |             |             |
> > > |-----|--------|------|-------|
> > > | Model             | Text            | Audio       | Group       |
> > > | Human             | 80.30/84.10     | 82.40/86.20 | 79.80/83.50 |
> > > | Random            | 25.0/21.0       | 25.0/22.0   | 16.67/15.33 |
> > > | CLAP              | 33.27/37.21     | 6.14/8.12   | 4.66/6.23   |
> > > | CLAP-LAION        | 34.78/38.43     | 6.52/8.94   | 5.07/7.82   |
> > > | CompA-CLAP (ours) | 44.28/46.34     | 22.52/24.31 | 15.13/17.10 |
> > >
> > > Table 2: Performance comparison of CompA-CLAP with baselines on Real-world / Synthetic distribution in
> > > CompA-attribute benchmark.
> > >
> > > >Q. The choice of metrics seems not entirely clear. It would be helpful to either (1) cite works which already use similar metrics, to motivate their used based on wide preexisting adoption in the ALM community, or (2) provide more detailed motivation for why the metrics are appropriate. For example, why is "group score" the conjunction of audio score and text score and not, e.g., their harmonic mean in the style of an F-score?
> > >
> > > **Ans:** As mentioned in Section 2.4, we are inspired by Winoground (Thrush et al., 2022) for our evaluation metrics. Winoground, on the other hand, is inspired by pioneering work on composition understanding in the Winograd schema challenge (Levesque et al., 2012). This has been widely used for a variety of language-related tasks (Rudinger et al., 2018; Sakaguchi et al., 2021; Zhao et al., 2018).
> > >
> > > Group score over Harmonic mean:
> > > The harmonic mean gives more weight to lower scores. In the context of retrieval tasks, we often want to ensure that both modalities perform well, and a poor performance in one should not be compensated by a high performance in the other. The group score, by considering both scores together, provides a more balanced view of the system's overall effectiveness.
> > >
> > > >Q. There are no quantifiable measurements of incertainty or variability in the comparisons presented in the paper. The authors say that their results are averaged over 3 random seeds; can they provide the (variance, stddev) over these seeds? What about Clopper-Pearson or other appropriate confidence intervals for the metrics reported? As is, it is impossible to assess the significance of the differences between metrics in Table 2.
> > >
> > > **Ans:** Thank you for your valuable feedback. We have added the standard deviation for our proposed models.
> > >
> > > >Q. Some important experimental design decisions in the paper are not clearly motivated and not validated with empirical studies. This includes:
> > > >- Why did the authors only fine-tune CLAP (when they have already shown the CLAP weights to be ineffective) instead of training it from scratch, when the authors apparently have access to the full original pretraining datasets?
> > >
> > > **Ans:** Thank you for your question. There are several reasons to do this and we will list down the individual reasons here:
> > > - We fine-tune CLAP to teach compositional reasoning. Compositional reasoning is a significantly more difficult task and CLAP could learn this better and faster if it already possesses knowledge about individual acoustic events. CLAP gains this knowledge from large-scale contrastive pre-training (the first stage, before hard-negative-based finetuning).
> > > - As explicitly mentioned in Section 3.1 of our paper, current training datasets are insufficient for learning compositional reasoning. This is due to the acute lack of compositional audio in them (and openly available). The primary requirement of hard-negative-based fine-tuning is the requirement for compositional audio. Additionally, audios in these benchmarks are not explicitly annotated for being compositional or not. Thus, training for compositionality from scratch on the entire pre-training dataset was not a feasible solution.
> > > Note: Though our proposed CompA-661k improves over these factors, but is still im-balanced (see details in Table X) . Thus even CompA-661k is not suitable for compositional fine-tuning from scratch.
> > > - Fine-tuning from a pre-trained checkpoint is a common practice for learning compositional reasoning. This has been followed by several prior-work

---

> ### Author Response · Authors · 2023-11-14
> **Response to Reviewer Xast (4/6)**
>
> >Q. Why do the authors fine-tune two separate times, instead of performing a single phase of joint fine-tuning?
>
> **Ans:** We implement a two-stage training strategy to address specific challenges observed in the fine-tuning process as pointed out by Goyal et al. (2022)[1], Wortsman et al. (2022)[2], and Kumar et al. (2022). The first stage involves fine-tuning the model while sampling a single positive and negative per audio. This initial phase focuses on handling hard negatives in a batch. The second stage follows, fine-tuning the model with multiple positive and negative audi- text pairs, addressing the complexity of having multiple positive and hard negative sentences simultaneously. This two-stage learning approach is designed to allow the model to first learn to handle hard negatives individually before tackling the more intricate task of managing multiple positive and hard negative instances concurrently. Empirically, we find 2 stage training to perform better than single stage training.
>
> >Q. Why does the fine-tuning occur (hard negatives (3.3) --> synthetic pairs (3.4)) instead of the reverse ordering?
>
> **Ans:** The core rationale behind performing hard negative training first and modular contrastive training next is a Coarse-to-Fine Contrastive Learning approach. Hard negative training with an entire caption as a positive or a negative in the batch is an easy task and teaches CLAP to distinguish between entirely different contexts and concepts. On the other hand, modular contrastive training, a bit more difficult task, makes CLAP focus on specific compositional relationships of the entire scene, thereby refining its ability to discern subtle distinctions and nuances within similar contexts.
> Table 3 provides results when the order is changed. Empirical results prove that the order matters for training.
>
> |                            | CompA-Order |       |       | CompA-attribute |       |       |
> |----------------------------|-------------|-------|-------|-----------------|-------|-------|
> | Model                      | Text        | Audio | Group | Text            | Audio | Group |
> | CompA-CLAP (ours)          | 40.70       | 35.60 | 33.85 | 44.28           | 22.52 | 15.13 |
> | CompA-CLAP (order changed) | 38.35       | 32.75 | 24.65 | 41.52           | 19.15 | 12.17 |
>
> Table 3: Comparison of CompA-CLAP results with the order of compositionally aware hard negatives and modular contrastive learning changed.
>
> >Q. All of these decisions could be validated empirically, but I do not see these results in the paper.
> >- Both examples in Figure 4 both seem to be flawed training example. (1- Left) The negatives both appear to be correct. All of these examples (the true caption, and the negatives) describe three sound events occurring simultaneously. (2 - right) "Tiger roars amidst thunker" and "Tiger roar followed by" cannot both be positive labels for the same sample - only one can be correct. Both of these issues seem to raise concerns about the quality of the data used here; particularly since the "highly custom prompt" described does not appear to be shared in the paper.
>
> **Ans:** We apologize for any confusion and would like to provide a more detailed clarification regarding Figure 6. The figure illustrates three distinct acoustic events: "Tiger roar," "Thunder," and "Human conversation." The ultimate objective is to combine these acoustic events to construct a coherent acoustic scene, specifically "Tiger roar followed by human conversation amidst thunder." As thunder serves as a background element, "Tiger roars amidst thunder" is considered a valid positive scenario. Furthermore, within the acoustic scene, the tiger roar precedes the human conversation, making "Tiger roar followed by human conversation" another valid positive sample. You can find more examples in Appendix, Table 9.
>
> >- The paper makes several additional changes to the CLAP model.
> **Ans:** Yes we only make a single change to the original CLAP model proposed by Wu et al. (2023). We replace the RoBERTa encoder with an instruction-tuned Flan-T5-large encoder (Chung et al., 2022) as this has shown to further improve performance. This is detailed in Section 3.2 Methodology paragraph.

---

> > ### Author Response · Authors · 2023-11-14
> > **Response to Reviewer Xast (5/6)**
> >
> > ### Minor Comments:
> >
> > >Q. "Sequence" and "attribute binding" should be defined clearly and early in the paper (ideally before these phrases are used).
> >
> > **Ans:** Thank You for your suggestion. We have decided to incorporate this in our Introduction section. Our Section 1, Paragraph 3, Main Contribution point 1 had the following line:
> >
> > Original: While CompA-order is used to evaluate the models’ ability to understand the order of occurrence between two acoustic events in an audio, CompA-attribute is used to evaluate the models’ ability to understand attribute-binding for acoustic events.
> > We have now changed this to:
> >
> > Modified: While CompA-order is used to evaluate the models’ ability to understand the order of occurrence between two acoustic events in an audio, CompA-attribute is used to evaluate the models’ ability to link attributes to specific acoustic events (attribute-binding).
> >
> > We have also made a similar change in our abstract. We have also removed the word sequence from Section 1, Paragraph 2 and replaced it with “order of occurrence”. Please find these changes in the revised version of the paper.
> >
> > >Q. Caption in Figure 2 should describe the metrics being presented in the Figure clearly. The Figure is also missing clear y-axis labels. The phrase "minor" is not quantifiable. This figure could also be improved by (a) measures of variability such as error bars or Clopper-Pearson confidence intervals or (b) baselines to show that the CLAP models actually degrade to "trivial" performance (is R@1 of 0.42 on AudioCaps val, or R@1 of 0.35 on AudioCaps test, "trivial"?).
> >
> > **Ans:**  Thank you for your suggestion, we have incorporated this change.
> >
> > >Q. A lot could be done to make the paper more self-contained. As-is, it is missing a great deal of relevant background information, which is exacerbated by the deferral of the related work to Section 5.
> >
> > **Ans:** We have incorporated several changes to the paper. We would request the reviewer to please go through the revised version. Additionally, deferring Section 5 to later sections in a common practice followed, and we believe it highlights our primary contributions better.
> >
> > >Q. "to assure high quality, we don’t concatenate or overlay random acoustic events but ask an LLM to create unique audio scenes based on the available labels" - please clarify both the overall procedure here, and how this assures high quality.
> >
> > **Ans:** Thank You for the question. As described in SubSection 3.4, our proposed modular contrastive learning approach is aided by synthetic data generation using a simple template-based synthetic audio-caption pair creation algorithm. This algorithm includes concatenating and/or overlaying audio snippets containing a single acoustic event, to make a final scene. However, concatenating random acoustic events can result in un-realistic scenes, thereby causing a shift from real-world audios that the model might generally see. Thus, we ask GPT to create scenes for us which can occur in the real-world. Following, we give 2 examples of un-realstic scenes generated randomly and 2 scenes created by GPT. We already provide a list of scenes created by GPT in Table 9 in the Appendix.
> >
> > Scenes created by GPT:
> >
> > (Tap dance + Applause) ∗ Female speech, woman speaking
> > Chicken, rooster ∗ (Environmental noise + Mosquito + Child speech, kid speaking)
> >
> > Random Scenes:
> > (Bird call + Drill) * Music
> > (Engine firing + Dog panting + Environmental noise) * Bell ringing
> > Here “+” refers to concetanation and “*” refers to overlay.
> >
> > >Q. "where augmenting LAION-audio-630K with 2 million audios from AudioSet improved performance on benchmarks only marginally." Please clarify what is meant in this sentence.
> >
> > **Ans:** Table 3 of Wu et al. (2023) shows that augmenting (or adding) LAIONAudio-630K with the entire 2M AudioSet for CLAP training almost never improves performance. Since AudioSet does not have gold captions (neither time-stamps for acoustic events) and only labels, the authors generate captions using Keyword-to-Caption Augmentation. However, this type of caption generation does not guarantee to capture temporal sequence correctly. Thus, we conclude that even using a large-scale dataset with temporally unaligned captions does not lead to performance improvement. Conversely, our experiments with CompA-661k show that adding a small amount of audio with temporally aligned captions improves scores by significant margins.
> >
> > >Q. The "experimental protocol" sections in 3.2 and 2.4 here do not describe protocols; please clarify or remove these sections. I would recommend a single "experimental protocol" section, since these subsections only describe individual components of the overall experimental setup.
> >
> > **Ans:** We have re-written experimental protocol in each of the subsections in 3.2, 3.3 and 3.4. However we believe that these sections are essential for reporoducibility and clear understanding of the training methodologies as each methodology has several hyper-parameters.

---

> > > ### Author Response · Authors · 2023-11-14
> > > **Response to Reviewer Xast (6/6)**
> > >
> > > ### Typos etc.
> > >
> > > >Q. Section 1 should reference that prior work is deferred to, and discussed in, Section 5.
> > >
> > > **Ans:** We have added this to our paper.
> > >
> > > >Q. "attribute-binding" vs "attribute binding" both used in the paper.
> > >
> > > **Ans:**  We have changed all instances of attribute binding to attribute-binding.
> > >
> > > >Q. "The group score evaluates of the model performed well for the evaluation instance in the benchmark" - this is not a meaningful description.
> > >
> > > **Ans:** We have added a more detailed description of the group score in our paper with proper citations, describing why the group score is important. Our group score is motivated by prior work by Elazar et al. (2021), who show that the individual evaluation metrics tend to overestimate model performance by computing scores for the twin sentences individually.
> > >
> > > >Q. 3.4: "due the the inherenly complex and non-linear nature of audios in the wild" - please clarify what is meant here, and how nonlinearity relates to the hardness of an example.
> > >
> > > **Ans:** A good example of a complex non-linear audio with multiple acoustic events is a recording of a bustling city street. In this audio, you might hear a variety of sounds overlapping and interacting in complex ways: the continuous hum of traffic, intermittent honking of car horns, the rhythmic footsteps of pedestrians, occasional snippets of conversation, the distant ringing of bicycle bells, and perhaps the sudden, sharp sound of a door slamming.
> > >
> > > The non-linear aspect in this example refers to the way these different sound sources interact with each other. Unlike linear sounds, where each component can be easily isolated and doesn't significantly affect or change due to the presence of other sounds, non-linear audio involves components that can interact in unpredictable ways. For instance, the sound of a car horn might be partially muffled by the rumble of a passing truck, or the clarity of a conversation might fluctuate due to the varying intensity of surrounding traffic noise. These interactions are not straightforwardly additive and can create a complex soundscape where the individual sources of sound influence each other in a dynamic, non-linear manner.
> > >
> > > Non-linear sounds make hard examples to learn from.

---

> ### Author Response · Authors · 2023-11-16
> **Request to review the rebuttal**
>
> Dear reviewer Xast,
>
> Thank you for the time you spent reviewing our paper. We have submitted our response to your concerns, including a revised version of our paper. Please let us know your comments and if you have any more concerns. Thank You again!

---

> > ### Comment · Reviewer_Xast · 2023-11-17
> > **Rebuttal follow up [1/2]**
> >
> > Thank you to the authors for the detailed response and the major edits made to incorporate the suggestions and questions from my initial review. The authors' response addressed many of my concerns (i.e. about tone of the paper, experimental details, empirical and experimental considerations) and provided some critical details which have improved the paper considerably.
> >
> > I am open to raising my score to the paper due to the improvements from these significant changes. I would also like to highlight some concerns that have not yet been addressed from my original review:
> >
> > * From original review: "There is no mention in the paper of where the benchmark can be accessed, how it can be used by other works, what format it is provided in, etc. This is essential information for a public benchmark (it is ok to maintain anonymity by describing where the benchmark will be available upon acceptance, if the description is sufficiently clear)."
> >
> >   * ^ this concern has not been addressed. The authors' response to this point above seems to be to an entirely different point, perhaps this is an oversight; if not, I do not understand how it addresses the original concern about how the benchmark will be shared or accessed. Again, I must emphasize how critical this is: if the authors are proposing a *benchmark*, that benchmark needs to be thoughtfully constructed for maximal ease of use, highly accessible, and (ideally) this would be verifiable to reviewers during this process. However, given the need ot preserve anonymity in the review process, I am also willing to accept a reasonable and realistic description of how the authors *plan to* make this benchmark available, provided there is sufficient detail (format, source, plan to ensure availability, documentation, etc.).
> >
> > * I acknowledge the authors' response to my concerns on dataset size. However, I still don't believe that this addresses the point. I certainly acknowledge the difficulty of constructing a human-annotated audio benchmark, but this does not change the statistical fact that obtaining precise estimates from the benchmark will not be possible, beyond a certain point, due to the sample size. However, I also acknowledge that the *quality* of the benchmark is perhaps even more important than size, beyond a certain point, and the dataset does appear to have high-quality annotations -- which the authors clarifications helped elucidate - thank you for the updated details in Section 2.
> >
> >   - Since the authors construct 3 random seeds to effectively compensate for the small size of the benchmark, perhaps they can also recommend a procedure to others to follow along these lines (in the supplement). This would ensure uniformity in how future researchers utilize the benchmark and help make results comparable across future works that utilize it.
> >
> >   - Related to the above point, the authors' response doesn't seem to address the fact that there are two separate benchmarks consisting of 400 and 200 samples each -- not a single benchmark of 600 samples. This matters a lot when it comes to measures of statistical certainty, which usually decay quadratically in variance with the test set size.
> >
> >   - Related to the authors' point about data scarcity: I do not feel that audio data is so scarce as to justify the small size. There are several massive video datasets (e.g. YT-8M) which include audio, and many large sound effects datasets (including all of LAION CLAP's training datasets, some containing hundreds of thousands of clips) which could be used as additional pools, particularly if single-source audio could be easily pseudolabeled and then summed to form multi-sound clips.
> >
> > * Real world vs. synthetic: Thank you for the clarification. This table should be included in the paper (I don't see it there?) - as it does seem to indicate that the synthetic data is systematically easier.
> >
> > * choice of metrics: Thank you for the clarification. Are these clarifications and citations in the paper?
> >
> > * Variability of performance metrics: These considerably strengthen the results; thank you for adding them to Table 2. Why are the metrics not reported for the remaining models and baselines, and is it possible to add them?
> >
> > * Fine-tuning details: Again, the author clarifications helped a great deal here. Please make sure these clarifications are included in the paper, as it was not at all obvious to me that this setup was the way to proceed here (even as a researcher familiar with the fine-tuning and foundation modeling literature). Again, "Table 2" from the author response should be in the paper (supplement is fine).

---

> > > ### Comment · Reviewer_Xast · 2023-11-17
> > > **2/2**
> > >
> > > * Thank you for the clarification and updates surrounding Figure 6. I am a bit confused about the annotations in the new figure. I see an unambiguous annotation of the form: A_1 = (A_1^1 + A_1^2)*A_1^3. It seems that the text label for this ("Tiger roar followed by human conversation amidst thunder") is ambiguous (is the thunder happening amidset the human conversation, or is the thunder happening amidst the human conversation AND the tiger roar). The formal label shows that "amidst" modifies both - can the authors offer any insight into how this linguistic ambiguity may affect either the data, or the model learned from it?
> > >
> > > * The "Overview & Background" sections are a nice addition and have improved the papers self-containedness a great deal.
> > >
> > > * The authors' clarification regarding my question "where augmenting LAION-audio-630K with 2 million audios from AudioSet improved performance on benchmarks only marginally" would be useful in the paper; I don't see a similar clarification there and would suggest to add it to improve clarity. (Indeed, many of the authors' helpful clarifications to my review don't seem to be in the paper, but I feel that they would improve its clarity and would suggest to add them, as space allows.)
> > >
> > > Again, thanks to the authors for their extensive and thorough response, it did improve the paper considerably.

---

> > > > ### Author Response · Authors · 2023-11-18
> > > > **Response to rebuttal follow-up (1/3)**
> > > >
> > > > Dear Reviewer Xast,
> > > >
> > > > Thank You for reading our rebuttal. We greatly appreciate the time you spent on reviewing our paper in detail. Your suggestions are of great help to improve the quality of our paper. Additionally, we are grateful to know we have clarified most of your questions. We have answered your additional questions and also uploaded a revised version. We would request you to go review it.
> > > >
> > > > >Q.From original review: "There is no mention in the paper of where the benchmark can be accessed, how it can be used by other works, what format it is provided in, etc. This is essential information for a public benchmark (it is ok to maintain anonymity by describing where the benchmark will be available upon acceptance, if the description is sufficiently clear)."
> > > >
> > > > >Q.Since the authors construct 3 random seeds to effectively compensate for the small size of the benchmark, perhaps they can also recommend a procedure to others to follow along these lines (in the supplement). This would ensure uniformity in how future researchers utilize the benchmark and help make results comparable across future works that utilize it.
> > > >
> > > > **Ans:** We would like to answer both these questions together as they are related.
> > > > First of all, we sincerely apologize for the mistake that caused the 1st question. It was indeed an oversight and happened while we were drafting different parts of the reply. However, we have edited the original answer, and we are also mentioning our response here for your convenience:
> > > >
> > > > Post acceptance, we plan to release the entire dataset on our GitHub with CC-BY-NC 4.0 License. Together with the benchmark, we also plan to release all evaluation codes and training codes for the benchmark with appropriate seed settings. All of this will be supported with proper documentation.
> > > >
> > > > To ensure uniformity in scores, we plan to release a website similar to the SUPERB Benchmark that is meant for spoken language processing tasks (https://superbbenchmark.org/). Researchers will be able upload their models for automatic evaluation and see how their models perform on private and public leaderboards. Since evaluation can be done on a CPU, this would not be a great overhead. The leaderboard will host scores by averaging across 3 runs and report standard deviation across scores.
> > > >
> > > > For now, we have provided a sample of our dataset in a presentation (.pptx file) in the supplementary materials and have also included an anonymous link to access the complete dataset in the supplementary readme file.
> > > >
> > > > >Q. Related to the above point, the authors' response doesn't seem to address the fact that there are two separate benchmarks consisting of 400 and 200 samples each -- not a single benchmark of 600 samples. This matters a lot when it comes to measures of statistical certainty, which usually decay quadratically in variance with the test set size.
> > > >
> > > > **Ans:** We apologize if you felt our response was not clear about this detail. The sentence “Our dataset has 200 more samples than theirs.” denoted the entire combined benchmark, and we agree this might have been confusing. However, we mention in both Sections 1 and 2 of our benchmark the actual size –  400 + 200. CompA-order is of the same size as Winoground (Thrush et al. ,2022). CompA-attribute is smaller. CompA-order, however, has triplets for 100 samples which Winoground does not.
> > > >
> > > > >Q. Real world vs. synthetic: Thank you for the clarification. This table should be included in the paper (I don't see it there?) - as it does seem to indicate that the synthetic data is systematically easier.
> > > >
> > > > **Ans:** Thank you for your question. Yes, this table is included this in the paper in the Appedix C.3 as Table 10.
> > > >
> > > > >Q. choice of metrics: Thank you for the clarification. Are these clarifications and citations in the paper?
> > > >
> > > > **Ans:** Yes. We have added these details to Section 2.4 with appropriate citations.
> > > >
> > > > >Q. Variability of performance metrics: These considerably strengthen the results; thank you for adding them to Table 2. Why are the metrics not reported for the remaining models and baselines, and is it possible to add them?
> > > >
> > > > **Ans:** Thank You for the suggestion. We misunderstood your prior message and assumed you had requested only for CompA-CLAP methods. We have just made a revision to the paper with metrics added for each baseline.
> > > >
> > > > >Q. Fine-tuning details: Again, the author clarifications helped a great deal here. Please make sure these clarifications are included in the paper, as it was not at all obvious to me that this setup was the way to proceed here (even as a researcher familiar with the fine-tuning and foundation modeling literature). Again, "Table 2" from the author response should be in the paper (supplement is fine).
> > > >
> > > > **Ans:** We are delighted to know that we have clarified your questions. We have added this detail to Paragraph 1 and 2 in Section 3.4 about the intuition behind our setup. Also, as mentioned above we have added Table 2 from our response as Table 10 in Appendix C.3.

---

> > > > > ### Author Response · Authors · 2023-11-18
> > > > > **Response to rebuttal follow-up (2/3)**
> > > > >
> > > > > >Q. Related to the authors' point about data scarcity: I do not feel that audio data is so scarce as to justify the small size. There are several massive video datasets (e.g. YT-8M) which include audio, and many large sound effects datasets (including all of LAION CLAP's training datasets, some containing hundreds of thousands of clips) which could be used as additional pools, particularly if single-source audio could be easily pseudolabeled and then summed to form multi-sound clips.
> > > > >
> > > > > **Ans:** We acknowledge the presence of additional datasets available, such as YT-8M and LAION Audio-630k. Prior to the start of the annotation process, we did consider these options. However, we would like to list the difficulties we faced:
> > > > >
> > > > > 1) A major problem with using any large-scale dataset for annotation is the lack of captions or temporally synchronized labels (like in AudioSet strong). Pseduolabeleing of audio events still does not provide us with temporally synchronized labels. Without either of these details, it is difficult for us to curate a high-quality dataset. Why? Here, we list 2 reasons:
> > > > >
> > > > >     - As a first step to reducing the search space for pairs,  we programmatically filter the AudioSet strong dataset with the temporally synchronized labels. Code can be found in Supplementary and details in Appendix B. 3. An example of a filter rule is: “find audio with a pair of acoustic events and then find another with the same events that are reversely ordered but allow a maximum of 2 extra events in between with a total duration less than 50% of the nearest event. Note that these specific rules were found after extensive qualitative studies. There filtered audios were then presented to the annotators in an annotation tool, found in Fig. 7 in the Appendix. Manually finding pairs and reverse pairs would result in a huge search space, which was not possible (n x n, where n is the size of the dataset).
> > > > >
> > > > > 	- Even filtering does not ensure high quality, which calls for expert annotation. Details about our annotation rules can be found in Section B3 (primarily points 1 and 2). Let us take an example of audio with 2 primary events: a tiger roaring followed by a human speaking. Similarly, let us consider an audio with the reverse order. For annotation, a strict rule is that the background sounds should not overpower or obscure the primary events, even after the rules mentioned in point a). Also, the intervening sounds (if any) in between the sounds should be natural and not very artificial (for example, there are examples in the AudioSet where music is intervened by a goat sound. Though this meets all our criteria, it is not a good naturalistic example.
> > > > >
> > > > > 2) Problem with LAION Audio-630k: Figure 4 in our paper shows that ~64% of LAION Audio-630k is composed of audios with single acoustic events. Out of the remaining ~36%, ~15% are from music datasets (MusicCaps, Musical Instrument) and are not useful for our benchmark. Finally, out of the remaining ~17% that consists of 107100 audios, post pre-processing, we rarely found the reverse of audios as these benchmarks were curated with diversity in mind.
> > > > >
> > > > > Examples of audios can be found in the PPT in our Supplementary. We also plan to expand CompA with more audios as part of future work, possibly temporarily labeling AudioSet 2M audios with GPT4-V.
> > > > >
> > > > >
> > > > > >Q. Thank you for the clarification and updates surrounding Figure 6. I am a bit confused about the annotations in the new figure. I see an unambiguous annotation of the form: A_1 = (A_1^1 + A_1^2)*A_1^3. It seems that the text label for this ("Tiger roar followed by human conversation amidst thunder") is ambiguous (is the thunder happening amidset the human conversation, or is the thunder happening amidst the human conversation AND the tiger roar). The formal label shows that "amidst" modifies both - can the authors offer any insight into how this linguistic ambiguity may affect either the data, or the model learned from it?
> > > > >
> > > > > **Ans:** Thanks for pointing this out. In Figure 6, we missed a comma in the caption for audio A_1. We have updated the diagram in the revised version of the paper. We have ensured that the comma is placed appropriately in the audio captions while training the model for unambiguous captions. For example:-
> > > > >
> > > > > A_1 = (A_1^1 + A_1^2)*A_1^3, Text: “Tiger roar followed by human conversation, amidst thunder”
> > > > > A_1 = A_1^1 + ( A_1^2 * A_1^3), Text: “Tiger roar, followed by human conversation amidst thunder”
> > > > >
> > > > > Code for the same can be found in our Supplementary metrical (post-neg-generation.ipynb).

---

> > > > > > ### Author Response · Authors · 2023-11-18
> > > > > > **Response to rebuttal follow-up (3/3)**
> > > > > >
> > > > > > >Q. The authors' clarification regarding my question "where augmenting LAION-audio-630K with 2 million audios from AudioSet improved performance on benchmarks only marginally" would be useful in the paper; I don't see a similar clarification there and would suggest to add it to improve clarity. (Indeed, many of the authors' helpful clarifications to my review don't seem to be in the paper, but I feel that they would improve its clarity and would suggest to add them, as space allows.)
> > > > > >
> > > > > > Thank You for this. Yes we have added this line to the revised version of our paper (Section 3.1, highlighted in blue). We would also like to mention that we have included details for all your suggestions in either the main paper or the Appendix. (Note: additions to the Appendix were not highlighted, but all additions to the main paper are highlighted in blue.) For better clarity, we have slightly updated some of our previous responses (from the initial rebuttal) to highlight the sections in which we have made the changes.
> > > > > > Below are the few changes that we have made in the revised version of the paper to incorporate your suggestions.:
> > > > > > 1) "only a word changed" (a few sentences after the claim "the audios have the exact same words") - Addressed in Section 2.2
> > > > > > 2) "it is trivial for ALMs to perform well on these benchmarks" - addressed in Section 2.1
> > > > > > 3) "CLAP undergoes only minor degradation in retrieval performance" - Addressed in the caption of Figure 1.
> > > > > > 4) "25% synthetic...but highly curated by expers" (what is highly curated?) - Addressed in Section 2.3
> > > > > > 5) "a trivial but ineffective solution" - Addressed in Section 3.1
> > > > > > 6) "a highly custom prompt" - Addressed in Section 2.3
> > > > > > 7) "Our version outperforms ... by significant margins" (no measures of significance used) - Addressed in Section 3.2
> > > > > > 8) We expand on details about the annotation in Section 2 and include more details about the exact annotation process.
> > > > > > 9) In Appendix B3, we provide more details about annotator demographics.
> > > > > > 10) In Appendix B3, we list down some rules adhered to during the annotation process.
> > > > > > 11) Added Table 10: Performance comparison of CompA-CLAP with baselines on Real-world / Synthetic distribution in CompA-attribute benchmark in Appendix C.3
> > > > > > 12) We have added the standard deviation for our proposed models and baselines used.
> > > > > > 13) We have re-written experimental protocol in each of the subsections in 3.2, 3.3 and 3.4.
> > > > > > 14) We have changed all instances of “attribute binding” to “attribute-binding”.
> > > > > > 15) We have added a more detailed description of the group score in Section 2.4.

---

> > > > > > > ### Comment · Reviewer_Xast · 2023-11-18
> > > > > > > **Increasing score to 6**
> > > > > > >
> > > > > > > Thank you to the authors for their detailed responses. I am going to increase my score to 6, as my main reservations have been addressed.

---

> > > > > > > > ### Author Response · Authors · 2023-11-18
> > > > > > > > **Thank You!**
> > > > > > > >
> > > > > > > > We are glad we could address your concerns. We thank you again for the detailed review and insightful comments, which helped us improve the quality of the paper.

---

### Official Review · Reviewer_jJm8 · 2023-10-30

**Soundness:** 4 excellent
**Presentation:** 3 good
**Contribution:** 4 excellent
**Rating:** 8
**Confidence:** 4

**Summary:**

This paper simultaneously propose new datasets (CompA) and a novel training method (CompA-CLAP) for the task of compositional reasoning in Audio Language Models (ALMs). The dataset contribution is separated between two subsets, CompA-order, which is targeted at the temporal ordering of audio events, and CompA-attribute, which targets attribute reasoning. Additionnally, the authors show that most ALM fail at grasping compositonal reasoning and propose large improvements on this problem with a new method, CompA-CLAP which both introduce a specific loss and a methodology for modular contrastive learning.

Overall, the paper is very well written and the methodology is very soundly presented by both exhibiting a profound issue of ALM, proposing novel datasets and even a large improvement in terms of learning methodology. My major (and almost only) criticism comes from the fact that the concept of « compositionality » presented on this paper is rather largely focused on « temporal » compositionality (even in the attribute case), and the paper would gain from a broader presentation of different possible types of compositionality. However, I still strongly believe that this paper would be a good addition to ICLR and recommend for acceptance.

**Strengths:**

The paper is very rich in new proposals, but all of these remain soundly theoretically grounded. Also, the proposal of new highly curated datasets are also an always-welcome addition for the community.

**Weaknesses:**

As previously stated, my major (and almost only) criticism comes from the fact that the concept of « compositionality » presented on this paper is rather largely focused on « temporal » compositionality. Although I understand that it is mandatory to make choices as this is still a very young research topic, I think the paper would gain in strength if a more clear definition of various types of compositionality would be presented.

**Questions:**

With regards to generating examples, although I understand that using prompts and GPT can perform a large array of tasks, your temporal task could be quite easily simulated (collating examples together, such as done with the Mixup augmentations). Could you quantify gains and differences with those more straightforward approaches ?

---

> ### Author Response · Authors · 2023-11-13
> **Response to Reviewer jJm8**
>
> We appreciate your acknowledgment of CompA's importance, clarity, novelty, effectiveness, and consistent performance gains. We're grateful for your recommendation to accept our paper. We have tried to address each of your concerns point by point.
>
> ### Weaknesses:
> >Q. As previously stated, my major (and almost only) criticism comes from the fact that the concept of « compositionality » presented on this paper is rather largely focused on « temporal » compositionality. Although I understand that it is mandatory to make choices as this is still a very young research topic, I think the paper would gain in strength if a more clear definition of various types of compositionality would be presented.
>
> **Ans:** Thank you for your valuable suggestion. We have added a subsection in the Appendix discussing different types of compositionality observed in audio. The same can be found in Appendix A.
>
> ### Questions:
> >Q. With regards to generating examples, although I understand that using prompts and GPT can perform a large array of tasks, your temporal task could be quite easily simulated (collating examples together, such as done with the Mixup augmentations). Could you quantify gains and differences with those more straightforward approaches ?
>
> **Ans:** Thank You for the question. Below, we have compared the results with traditional text augmentation methods, such as switching nouns and verbs using the Spacy library. We find that GPT-generated negatives outperform traditional text augmentation techniques. We hypothesize that this is due to the lower-quality and incoherent negatives generated by traditional NLP methods. We compare negatives from various LLMs and traditional NLP methods in Table 8 in the Appendix. Additionally, we provide code in the Supplementary material.
>
> |                       | CompA-Order |       |       | CompA-attribute |       |       |
> |-----------------------|-------------|-------|-------|-----------------|-------|-------|
> | Model                 | Text        | Audio | Group | Text            | Audio | Group |
> | CompA-CLAP (GPT-4 - ours)     | 40.70       | 35.60 | 33.85 | 44.28           | 22.52 | 15.13 |
> | CompA-CLAP (Traditional NLP) | 35.25       | 30.45 | 17.80 | 39.37           | 17.26 | 12.54 |
>
> Table 1: Comparison of CompA-CLAP with negatives generated by GPT-4 and Traditional NLP methods.

---

### Official Review · Reviewer_x6Uk · 2023-11-08

**Soundness:** 3 good
**Presentation:** 2 fair
**Contribution:** 2 fair
**Rating:** 6
**Confidence:** 4

**Summary:**

Objects in an image have spatial relationships (which occur where in the image) where as events in an audio clip have temporal and attribute relationships (which sound occur when, who made which sound). The authors claim that the current audio-language models (ALMs) poorly model such temporal relationships. They propose a novel benchmark, "CompA" to test compositional reasoning (temporal as well as attribute relationships) abilities of ALMs. CompA is comprised of two parts. CompA-order to evaluate temporal relationship and CompA-attribute to evaluate attribute binding. Finally, authors introduce a model (CompA-CLAP) to improve ALM compositional reasoning abilities. CompA-CLAP is a fine-tuned CLAP model, where in the authors include compositionally aware hard-negatives for contrastive learning.These hard-negatives are derived using text-LLMs for the test set and template-based methods for the training set.

**Strengths:**

* The submission identifies a gap in current audio-language model evaluation i.e., temporal and attribute relationships of audio events, and proposed the novel CompA benchmark to evaluate the same.
* Introduced a novel CompA-CLAP model which learns audio event relationships through contrastive learning and goal motivated negative sample design.
* Well furnished details in appendices for reproducibility.

**Weaknesses:**

* The core technical contribution of submission boils down to - "selecting appropriate negatives for the task at hand", an established idea in contrastive learning, thus raising the question of novelty in terms of technical contribution.
* The structuring of manuscript could be better. e.g.,
  - Transition from section 3.3 to 3.4 is not clear in the first reading.
  - Every section reads like a mini-paper with it's own background, methodology. May be add a paragraph at the beginning of section to prep the reader of upcoming sections and their differences.

**Questions:**

* Have the author's considered rich transcription approach? Since authors are using strongly annotated data, can the model be trained to output a sequence like ..  <event1_start> <event2_start><event2_end><event1_end> .. here event2 occurs in the midst of event1. This would avoid the process of carefully designing negatives and the model would still learn temporal relationships among audio events. Attributes can also be modeled in a similar fashion. I would imagine this format could be more flexible in terms of attribute modeling. (e.g., personA-talking vs personA-singing and so on.. ). An LLM should have no problem taking such an output sequence and convert it to human-readable format (either implicitly or explicitly).
* Can the author's clarify if the order of hard negative training and modular contrastive training matter. (section 4: 2nd paragraph).

---

> ### Author Response · Authors · 2023-11-13
> **Response to Reviewer x6Uk (1/2)**
>
> We thank you for your thorough review and constructive feedback. We have tried to address each of your concerns point by point.
>
> ### Weaknesses:
> >Q. The core technical contribution of submission boils down to - "selecting appropriate negatives for the task at hand", an established idea in contrastive learning, thus raising the question of novelty in terms of technical contribution.
>
> **Ans:** We agree that hard-negative-based contrastive learning is a well-established idea in contrastive learning. However, as also stated in our paper, we would like to argue that just adding hard negatives is not our primary technical contribution. Our primary contributions are as follows:
>
> 1. (Technical) Section 3.3: We modify the formulation of naive contrastive learning with hard negatives such that hard negative captions for a particular audio are ignored by other audios in the batch.
> 2. (Technical) Section 3.4: We propose an entirely new algorithm called “modular contrastive learning.” Modular Contrastive Learning has two primary innovations:
>     - We propose a novel and simple methodology of data generation that overcomes the data scarcity issue of compositional audios: Template-based synthetic creation of audio-caption pairs.
>     - Our proposed method promotes fine-grained composition learning. Each audio in the batch has multiple positives that describe compositional relationships between certain acoustic events in the batch. Each audio in the batch also has multiple negatives for each positive. An example of this can be seen in Figure 4 (Figure 6 in revised version).
>     - We modify the contrastive learning formulation such that each audio in the batch ignores the positives and negatives for all other audio.
> 3. (Non-Technical) Section 2: We propose two benchmarks, CompA-order and CompA-attribute, to evaluate compositional reasoning in Audio-Language Models (ALMs). The expert-annotated benchmarks, which are also the first of their kind, are complex to curate and annotate and annotate and represent a significant advancement in an understudied field. By focusing on this niche, our benchmark paves the way for a more nuanced understanding and development of ALMs.
>
> > Q. The structuring of manuscript could be better. e.g.,
> >- Transition from section 3.3 to 3.4 is not clear in the first reading.
> >- Every section reads like a mini-paper with its own background, methodology. May be add a paragraph at the beginning of section to prep the reader of upcoming sections and their differences.
>
> **Ans.:** Thank you for bringing these concerns to our attention! In response, we've incorporated a few introductory lines at the beginning of each section to concisely outline the main theme and facilitate smoother transitions between sections, all while considering space constraints.

---

> ### Author Response · Authors · 2023-11-13
> **Response to Reviewer x6Uk (2/2)**
>
> ### Questions:
> >Q. Have the author's considered rich transcription approach? Since authors are using strongly annotated data, can the model be trained to output a sequence like .. <event1_start> <event2_start><event2_end><event1_end> .. here event2 occurs in the midst of event1. This would avoid the process of carefully designing negatives, and the model would still learn temporal relationships among audio events. Attributes can also be modeled in a similar fashion. I would imagine this format could be more flexible in terms of attribute modeling. (e.g., personA-talking vs personA-singing and so on.. ). An LLM should have no problem taking such an output sequence and convert it to human-readable format (either implicitly or explicitly).
>
> **Ans:** Yes, we have. Indeed, this is the exact methodology we employ to generate AudioSet-CompA, a novel dataset contribution of our paper (described in Section 3.3). As the Section specifies, we generate AudioSet-CompA by prompting GPT-4 for time-aligned compositional captions for audios in the AudioSet strong dataset. The AudioSet strong dataset has time-aligned labels, i.e., the start and the end time for each acoustic event in the 10-second audio is specified. An example is:” [('Male speech, man speaking', 2.532, 3.207),('Cluck', 3.087, 3.388),('Generic impact sounds', 3.181, 3.388)]” Thus, we prompt GPT-4 with these time-aligned labels and ask it to generate a coherent caption from this, and indeed, GPT-4 performs accurately at this task. The prompt is provided in Appendix B.5.
>
> For compositionally aware hard harmful generation, detailed in Section 3.3, we do not employ this as we already have captions available for each audio. Thus, we directly prompt GPT-4 to generate hard negatives from this caption. The prompt is provided in Appendix B.5.
>
> >Q. Can the author's clarify if the order of hard negative training and modular contrastive training matter. (section 4: 2nd paragraph).
>
> **Ans:** The core rationale behind performing hard negative training first and modular contrastive training next is a Coarse-to-Fine Contrastive Learning approach. Hard negative training with an entire caption as a positive or a negative in the batch is an easy task and teaches CLAP to distinguish between entirely different contexts and concepts. On the other hand, modular contrastive training, a bit more difficult task, makes CLAP focus on specific compositional relationships of the entire scene, thereby refining its ability to discern subtle distinctions and nuances within similar contexts.
>
> Table 1 provides results when the order is changed. Empirical results prove that the order matters for training.
>
> |                            | CompA-Order |       |       | CompA-attribute |       |       |
> |----------------------------|-------------|-------|-------|-----------------|-------|-------|
> | Model                      | Text        | Audio | Group | Text            | Audio | Group |
> | CompA-CLAP (ours)          | 40.70       | 35.60 | 33.85 | 44.28           | 22.52 | 15.13 |
> | CompA-CLAP (order changed) | 38.35       | 32.75 | 24.65 | 41.52           | 19.15 | 12.17 |
>
> Table 1: Comparison of CompA-CLAP results with the order of compositionally aware hard negatives and modular contrastive learning changed.

---

> > ### Comment · Reviewer_x6Uk · 2023-11-22
> >
> > Thanks to authors for providing detailed responses. Provided Table1 and associated explanation clarifies the importance of ordering. As for the former question of training the model to output a sequence like  <event1_start> <event2_start><event2_end><event1_end>, my question was not wrt to data generation. It was related to training paradigm itself. If the aim is to make the model learn temporal and attribute relationships, can it be done by making the model output a sequence line  <event1_start> <event2_start><event2_end><event1_end> instead of the approach presented in the paper.

---

> > > ### Author Response · Authors · 2023-11-23
> > > **Request to review the rebuttal**
> > >
> > > Dear Reviewer x6Uk,
> > >
> > > With the rebuttal window closing soon (in a few hours), we request you review our answer to your latest question (answered in Reply to Official Comment by Reviewer x6Uk) and let us know if you have further questions. We hope that addressing all your concerns can impact our score positively.
> > >
> > > Thank You again for your time.

---

> ### Author Response · Authors · 2023-11-16
> **Request to review the rebuttal**
>
> Dear reviewer x6Uk,
>
> Thank You for your time in reviewing our paper. We have submitted our response to your concerns, including a revised version of our paper. Please let us know your comments and if you have any more concerns. Thank You again!

---

> > ### Author Response · Authors · 2023-11-18
> > **Request to review the rebuttal**
> >
> > Dear reviewer x6Uk,
> >
> > Thank you for the time you spent reviewing our paper. We have submitted our response to your concerns, including a revised version of our paper incorporating all changes requested. Please let us know your comments and if you have any more concerns. Thank You again!

---

> > > ### Author Response · Authors · 2023-11-20
> > > **Request to review the rebuttal**
> > >
> > > Dear reviewer x6Uk,
> > >
> > > Thank you for the time you spent reviewing our paper. With the discussion period ending soon (22nd November), we would request you to please review our rebuttal and the revised version of our paper which addresses all your concerns.
> > >
> > > Please let us know your comments and if you have any more concerns. Thank You again!

---

> > > > ### Author Response · Authors · 2023-11-22
> > > > **Request to review the rebuttal**
> > > >
> > > > Dear reviewer x6Uk,
> > > >
> > > > Since today is the last day for author discussion, we request you to please review our rebuttal and the revised version of our paper, which addresses all your concerns.
> > > >
> > > > Please let us know your comments and if you have any more concerns. Thank You again!

---

> ### Author Response · Authors · 2023-11-22
> **Reply to Official Comment by Reviewer x6Uk**
>
> Thank You for the question. We are grateful for your time and glad we could address your concerns regarding the paper. We hope that addressing all your concerns can impact our score positively.
>
> We acknowledge that the suggested rich transcription methodology can be used for training CLAP. However, below, we would like to answer why we did not try the rich transcription methodology in our experiments. However, we assure you that we will add this experiment to the final version of our paper as requested.
>
> 1) CLAP-like models (like the one employed in experiments in our paper) are trained to understand natural language. Understanding natural language helps in free-form text-to-audio retrieval and vice-versa, which is used in downstream applications. Outputting such a sequence and training the model on such a sequence using contrastive learning affects natural language understanding of the model and might make the model act as a “bag of words”. Let's take an example:
>
>     Rich Transcription: <human> <bird></bird></human>
>
>     Two compositionally different captions:
>      - A person speaking in a serene outdoor setting, intermittently interrupted by the chirping of birds."
>      - Birdsong dominates the soundscape with faint, distant human voices in the background.
>
>     Rich transcriptions like the one provided can translate to compositionally different audios by highlighting the unique combination and interaction of various sound elements within the scene. Thus, this would hinder learning compositionality and specific nuances in audio.
>
> 2) Training using rich transcriptions avoids the model learning variabilities in free-form natural language. This hinders real-world downstream applications.
> 3) Rich transcriptions, though beneficial in learning temporal ordering, do not provide a straightforward way to capture noun attributes. Natural language captions often describe important nuances in audios through attributes binded with nouns, such as “sad” in “sad song” or “baby” in “baby tiger”. Thus, training with hard negative natural captions makes CLAP understand various audio cues, and training with rich transcriptions might make CLAP overfit to only focusing on temporal ordering.
> 4) Training this model would need an extra step in inference time to convert natural language into the rich transcription format since the model was trained to understand only rich transcriptions.
> 5) Elaborating on our motivation for Modular Contrastive Learning (Section 3.4) - Real-world audio is often complex and unpredictable with multiple acoustic events. This hinders the model from learning fine-grained attribute-binding and order information with captions or rich transcriptions. Thus, though rich transcriptions can be an alternative to compositionally-aware hard negative training (Section 3.3), Modular Contrastive Learning still solves this significant issue that rich transcription methodology still faces.
>
> We would also like to mention that our proposed approach for hard-negative-based contrastive learning, which employs compositionally different hard negative captions as negatives, has also been implemented by several prior works in the vision-language space [1,2,3]. All these works have also been cited in our paper. However, our exact formulation for loss calculation is different and innovative and is elaborated in the rebuttal - Response to Reviewer x6Uk (1/2).
>
> [1] Paiss, Roni, et al. "Teaching clip to count to ten." arXiv preprint arXiv:2302.12066 (2023).
>
> [2] Yuksekgonul, Mert, et al. "When and Why Vision-Language Models Behave like Bags-Of-Words, and What to Do About It?." The Eleventh International Conference on Learning Representations. 2022.
>
> [3] Momeni, Liliane, et al. "Verbs in action: Improving verb understanding in video-language models." Proceedings of the IEEE/CVF International Conference on Computer Vision. 2023.

---

> > ### Comment · Reviewer_x6Uk · 2023-11-23
> >
> > Although there are ways to bind attributes to nouns even in rich transcription, overall, i agree with your arguments (especially 1 & 2). Thank you for additional clarification. Changed my final score.

---

> > > ### Author Response · Authors · 2023-11-23
> > > **Thank You!**
> > >
> > > We are glad we could address your concerns. We thank you again for the detailed review and insightful comments, which helped us improve the quality of the paper.

---

### Author Response · Authors · 2023-11-14
**General Comment and Revised Version**

We thank the reviewers for their insightful and positive feedback! We are encouraged to find that the CompA benchmark will be a valuable resource to the audio community and is a significant step in the development of ALMs with advanced compositional reasoning abilities (Reviewers jJm8 and Cw3v). Additionally, we are happy to know that all reviewers found our methodology for fine-tuning CLAP for compositional reasoning, i.e., CompA-CLAP, sound. Finally, Reviewers x6Uk, Xast and Cw3v also find our contributions well-motivated and our qualitative and quantitative comparisons sufficient.

One common concern was the structuring of the paper, and the flow from one section to the other was not smooth at times. Additionally, some information that was left in the Appendix was requested to move to the main paper. To improve these aspects, we have incorporated all changes suggested by the reviewers and uploaded a revised version. We have also highlighted some key points in the paper, but overall, significant changes were made to the structure to make it easier to read and follow. We would like to request the reviewers to please go through the revised version, which addresses all presentation concerns.

Additionally, we will open-source our benchmark and code with proper documentation upon acceptance. For benchmark examples, we request the reviewers refer to the PPT provided in the Supplementary material. Currently, the entire code and benchmark can also be found in the Supplementary, where a README file details the contents.

---

### Meta-Review · Area_Chair_tr1S · 2023-12-06

**Metareview:**

Develops a framework for incorporating compositional reasoning in audio LMs (ALM). The paper presents 2 new benchmarks: CompA-order (order of events matter) and CompA-attribute (attribute of the event matter). The authors also propose CompA-CLAP, fine-tuned CLAP with a novel learning strategy focused on improved contrastive loss formulation to do well on the compositional tasks. The authors have promised to release the benchmarks post acceptance.

The reviewers commented that the submission identifies the gap in current ALM, proposes novel benchmarks and solutions to address it. The proposed solution, which is based on improving the negatives and positives used for contrastive loss may be of limited novelty. But the authors argue that they also improve the contrastive loss formulation. There were concerns about the overall organization of the paper, which the authors have sufficiently addressed during the rebuttal.

The paper went through a very thorough discussion phase, where the authors carefully addressed a lot of the concerns raised during the review and revised the paper. For example, rich transcription was floated as an alternative; the authors addressed why the proposed strategy has advantages over such an approach. The reviewers also had questions about the stagewise training strategy used by the authors and its usefulness. The authors presented ablations addressing the concerns.

Overall, the paper provides a useful benchmark and interesting ideas to improve the quality of ALMs.

**Justification For Why Not Higher Score:**

The paper presents a new benchmark and relatively novel ideas on how to improve ALMs. Some concerns do remain like the size of the benchmark (the authors do clarify the reason for this).

**Justification For Why Not Lower Score:**

All reviewers agree that the paper identifies an important limitation of current ALMs and provides benchmarks and solutions to address it. This is worth sharing with the community.

---

### Decision · Program_Chairs · 2024-01-16

Accept (poster)